# 1 A comprehensive review of tropospheric background

- ozone: Definitions, estimation methods, and meta-analysis
- of its spatiotemporal distribution in China
- Chujun Chen<sup>1</sup>, Weihua Chen<sup>1\*</sup>, Linhao Guo<sup>1</sup>, Yongkang Wu<sup>1</sup>, Xianzhong Duan<sup>1</sup>,
- Xuemei Wang<sup>1</sup>, Min Shao<sup>1</sup>
- <sup>1</sup>College of Environmental and Climate, Jinan University, Guangzhou, 510632, P. R. China
- \*Correspondence: Weihua Chen (chenwh26@jnu.edu.cn)
- Abstract. Background ozone (O<sub>3</sub>) represents the baseline concentrations in the absence of local
- anthropogenic emissions and is critical for understanding and mitigating tropospheric O<sub>3</sub> pollution.
- Accurate estimation of background O<sub>3</sub> constrains the maximum achievable benefits of precursor
- emissions control and informs effective air quality management. This review synthesizes the definition
- and estimation methods for background O<sub>3</sub>, including in situ measurement, statistical analysis, numerical
- modeling, and integrated methods. A meta-analysis of background O<sub>3</sub> in China from 1994 to 2020 reveals
- pronounced spatial variability, with concentrations ranging from 33 ppb in the Northeast China to 48 ppb
- in the Northwest China, and a national mean of 41 ppb, accounting for 79% of the tropospheric maximum
- daily 8-hour average O<sub>3</sub>. Methodological discrepancies are evident for background O<sub>3</sub>: in situ and
- 17 statistical methods yield higher estimates, whereas integrated approaches produce lower yet more
- 18 consistent values. Placed in a global context, background O<sub>3</sub> levels in China are medium-to-high and
- exhibit an increasing trend. This review highlights the need for integrated estimation methods to improve
- accuracy, underscores the international collaboration to address long-range pollutant transport, and calls
- 21 for further research on the interactions between background O<sub>3</sub> and climate change. By advancing the
- 22 understanding of background O<sub>3</sub> dynamics, it provides critical insights for atmospheric chemistry and air
- pollution control in China and beyond.

## 1 Introduction

- Since the implementation of the "Air Pollution Prevention and Control Action Plan" in 2013 and the
- subsequent "Three-Year Action Plan for Winning the Blue Sky War", China has achieved remarkable

improvement in air quality, particularly in reducing fine particulate matter (PM2.5) concentrations. Nationwide PM<sub>2.5</sub> levels declined by approximately 50% between 2013 and 2020 (Geng et al., 2024). However, surface ozone (O<sub>3</sub>) pollution has emerged as a growing concern. From 2015 to 2022, the number of O<sub>3</sub> pollution days increased steadily in major cities such as Beijing, Shanghai, and Guangzhou (Li et al., 2019; Wang et al., 2023), with exceedance days more than doubling in some regions (Ozone Pollution Control Committee of Chinese Society of Environmental Sciences, 2024). In response, the "Opinions on Deepening the Fight Against Pollution", issued by the Central Committee of the Communist Party of China and the State Council, incorporated coordinated control of both PM<sub>2.5</sub> and O<sub>3</sub> into the "14th Five-Year Plan" (2021-2025), marking a strategic shift toward multi-pollutant management. Tropospheric O<sub>3</sub> is a secondary pollutant formed through photochemical reactions involving volatile organic compounds (VOCs) and nitrogen oxides (NO<sub>x</sub>). It consists of two components: locally produced O<sub>3</sub> from anthropogenic emissions and background O<sub>3</sub>, both of which impact human health, ecological ecosystems, and agricultural productivity (McDonald-Buller et al., 2011; Wang et al., 2009b). Background O<sub>3</sub> refers to the O<sub>3</sub> concentration present in the absence of local anthropogenic precursor emissions. It originates from a variety of natural and non-local processes, including methane (CH<sub>4</sub>) oxidation, stratosphere-troposphere exchange (STE), vegetation, soil, lightning, wildfires and longrange pollutants transport, as shown in Fig. 1 (Dolwick et al., 2015; Thompson, 2019). Among these sources, CH<sub>4</sub> is unique due to its long atmospheric lifetime (8-9 years) and globally well-mixed distribution (Fiore et al., 2002b; Vingarzan, 2004; West and Fiore, 2005; Thompson, 2019). In the preindustrial era (~1750), natural CH<sub>4</sub> sources (e.g., wetlands, inland waters, geological emissions) contributed ~95% of global emissions (Lassey et al., 2000; Prather et al., 2012; Valdes et al., 2005), making CH<sub>4</sub> oxidation a stable background contributor to tropospheric O<sub>3</sub> (Skipper et al., 2021; Sun et al., 2024; Thompson, 2019; Vingarzan, 2004; Wu et al., 2008). However, over the past century, anthropogenic CH<sub>4</sub> emissions from agriculture, fossil fuels, and waste have increased substantially, accounting for over 60% of global CH<sub>4</sub> by 2012 (Fiore et al., 2002b; Jackson et al., 2024; Kirschke et al., 2013; Lelieveld et al., 1998; Saunois et al., 2016). This transition challenges the traditional classification

of CH<sub>4</sub>-driven O<sub>3</sub> as part of the "background," and calls for a more precise attribution that excludes anthropogenic CH<sub>4</sub> from background O<sub>3</sub> estimates. Differentiating natural from anthropogenic CH<sub>4</sub> contributions is thus crucial for improving the accuracy and policy relevance of background O<sub>3</sub> assessments.

Background O<sub>3</sub> typically contributes 60–80% of total tropospheric O<sub>3</sub> at both global and regional scales (Akimoto et al., 2015; Chen et al., 2022; Dolwick et al., 2015; Lee and Park, 2022; Lefohn et al., 2014; Zhang et al., 2011). Unlike PM<sub>2.5</sub>, which can be more directly mitigated through local emission reductions, O<sub>3</sub> management is more complex due to its nonlinearity and the presence of a large, unmodifiable background component (Chen et al., 2022). Although emission control measures in Europe, the United States (U.S.), and Japan have reduced O<sub>3</sub> exceedances, background O<sub>3</sub> levels have continued to rise (Akimoto et al., 2015; Cooper et al., 2012; Wilson et al., 2012; Yan et al., 2021). For example, in the U.S., the relative contribution of background O<sub>3</sub> to total ground-level O<sub>3</sub> has risen by approximately 6% over the past two decades (Jaffe et al., 2018), driven by climate change, rising CH<sub>4</sub>, and transboundary pollution (Chen et al., 2022; Vingarzan, 2004). This trend is especially concerning in East Asia, where background O<sub>3</sub> is further elevated by regional transport from neighboring countries, exacerbating the problem (Vingarzan, 2004). As anthropogenic NO<sub>x</sub> and VOCs emissions decline under stricter regulations, the relative importance of background O<sub>3</sub> in shaping observed pollution will only increase (Jaffe et al., 2018; Lam and Cheung, 2022; Skipper et al., 2021), limiting the effectiveness of local controls and demanding new international and multi-scale mitigation strategies.

Background O<sub>3</sub> defines the "baseline" for regional O<sub>3</sub> pollution and sets the upper bound of achievable air quality improvements through domestic anthropogenic emission controls (Fiore et al., 2014; Wang et al., 2009a; Zhang et al., 2011). Persistently high background O<sub>3</sub> levels complicate compliance with current and future O<sub>3</sub> standards, a challenge recognized by major agencies such as the National Aeronautics and Space Administration (NASA) (Huang et al., 2015; Thompson, 2019; Vingarzan, 2004) and in China's "Blue Book on Prevention and Control of Atmospheric Ozone Pollution" (Ozone Pollution Control Committee of Chinese Society of Environmental Sciences, 2022). Despite its importance, understanding of background O<sub>3</sub> remains constrained by inconsistent definitions, diverse

estimation methods, and limited regional assessments. While several recent reviews have touched on background O<sub>3</sub> in the broader context of O<sub>3</sub> pollution (Lu et al., 2019b; Liu et al., 2020; Sahu et al., 2021; Wang et al., 2022b; Xu, 2021), and a few studies have provided quantitative estimates and source apportionment of background O<sub>3</sub> in China (Sahu et al., 2021; Wang et al., 2022b; Chen et al., 2022; Wang et al., 2011), no dedicated, methodologically focused synthesis is available.

 This study addresses that gap by: (i) systematically reviewing the evolution of the background O<sub>3</sub> concept; (ii) providing a comparative assessment of the methods (i.e., in situ measurement, statistical analysis, numerical modeling, and integrated methods) employed to estimate background O<sub>3</sub>; and (iii) synthesizing the spatiotemporal patterns of background O<sub>3</sub> in China within a broader global context, examining the spatial and temporal variations of background O<sub>3</sub> across China using publicly available data. Finally, we identify key knowledge gaps and propose future research priorities to advance understanding and inform effective policy responses. As background O<sub>3</sub> increasingly shapes the O<sub>3</sub> landscape, this review provides timely insight into the scientific and regulatory frontiers of O<sub>3</sub> pollution control both in China and globally.

Figure 1: Conceptual diagram of tropospheric O<sub>3</sub> components and sources.

#### 2 Materials and methods

#### 2.1 Data source and study area

To provide a comprehensive synthesis of advancements in the study of background O<sub>3</sub>, a systematic literature search was conducted across major academic databases, including the Web of Science, Google Scholar, Science Direct (Elsevier), Scopus, Springer, Wiley, and China National Knowledge Infrastructure (CNKI). The search was centered on the following key thematic terms: background/baseline/natural ozone/O<sub>3</sub>, regional background ozone/O<sub>3</sub>, and policy relevant background ozone/O<sub>3</sub>, ensuring the inclusion of a wide range of relevant studies. This study identified 153 pertinent documents, comprising 132 peer-reviewed English-language papers, 10 peer-reviewed Chinese-language papers, 6 English-language reports, 1 English-language books, 2 Chinese-language books, and 2 Chinese-language master's theses. These documents form the core foundation of this review, which traces the evolution of the definition and estimation method for background O<sub>3</sub> over a span of seven decades (1952–2024), providing a comprehensive historical perspective on the development of the field. In addition to reviewing the definition and estimation method for background O<sub>3</sub>, we also analyzed the spatial and temporal characteristics of regional background O<sub>3</sub> concentrations in China during the period 1994-2020. This analysis was based on 44 peer-reviewed papers, including 28 papers in English and 16 papers in Chinese, which collectively provided over 700 data points on background O<sub>3</sub> concentration from various regions and time periods within China. To ensure the reliability and comparability of compiled background O<sub>3</sub> estimates, we adopted strict inclusion criteria: (i) reported values must specify temporal coverage and measurement or estimation methods; (ii) study location must be clearly identifiable within China; and (iii) data must originate from peer-reviewed publications, official reports, or other authoritative sources. All values considered in this study had already undergone rigorous quality control and screening by the original authors; therefore, no further outlier removal was performed. Where necessary, datasets were harmonized through standardized unit conversions and consistent temporal categorization to enhance comparability. Based on these criteria and harmonization procedures, the final dataset integrates estimates across

diverse temporal resolutions: annual data (31%, 235 data points), seasonal data (26%, 195 data points),

(March–May; 24%, 127 data points), summer (June–August; 28%, 145 data points), autumn (September–
November; 24%, 125 data points), and winter (December–February; 24%, 124 data points). A detailed
breakdown of regional and temporal distributions is provided in Table S1. In addition, maximum daily

8-hour average (MDA8) O<sub>3</sub> concentrations for the seven regions during 2013 to 2018 were obtained from

and monthly data (43%, 326 data points). The seasonal and monthly data were classified into spring

He et al. (2023).

To assess the regional differences in background O<sub>3</sub> concentrations across China, the country was categorized into seven geo-administrative regions based on a combination of social, natural, economic, and human environmental factors (He et al., 2023). These regions include Northeast China (NEC), North China (NC), East China (EC), Central China (CC), Northwest China (NWC), Southwest China (SWC), and South China (SC), as shown in Fig. 4. A detailed description of these regional divisions is provided in Table S2.

#### 2.2 Data process

The background O<sub>3</sub> concentrations presented in this study are reported as volume mixing ratios in parts per billion (ppb). In some studies, however, values are expressed as mass concentration (μg m<sup>-3</sup>). To ensure consistency with international standards and comparability with global datasets, unit conversions were performed using Eq. (1):

$$ppb = \left(\frac{24.5 \text{ L mol}^{-1}}{48 \text{ g mol}^{-1}}\right) \times (\mu \text{g m}^{-3}),$$
 (1)

Where 48 g mol<sup>-1</sup> is the molar mass of O<sub>3</sub> and 24.5 L mol<sup>-1</sup> is the molar volume of an ideal gas under the reference conditions of 25 °C and 1013.25 hPa, as specified in the 2018 amendment to China's Ambient Air Quality Standards (GB 3095–2012) issued by the Ministry of Ecology and Environment (https://www.mee.gov.cn/gkml/sthjbgw/sthjbgg/201808/t20180815\_451398.htm). These reference conditions are consistent with international practices, such as 25 °C in the U.S. (U.S. EPA, 2011), 20 °C in the European Union (European Parliament and Council, 2008) and better reflect typical meteorological conditions across most regions of China.

## 2.3 Trend analysis

This study employed linear regression analysis to examine the annual trend in background O<sub>3</sub> concentration and assess the statistical significance of these trends over time. Specifically, linear regression was applied to the mean background O<sub>3</sub> concentrations (derived from scatter plot data) across different years, using the least squares method to determine the relationship between background O<sub>3</sub> concentration and time.

To evaluate the model's performance, the coefficient of determination (R<sup>2</sup>) was calculated. R<sup>2</sup> represents the proportion of variance in background O<sub>3</sub> concentration explained by the linear model, indicating how well the model fits the observed data. Higher R<sup>2</sup> values suggest a strong fit, while lower values indicate a weaker fit. The P-value was also calculated to test the statistical significance of the linear relationship between background O<sub>3</sub> concentration and time. A smaller P-value (typically less than 0.05) indicates a statistically significant linear relationship, suggesting that the observed trend is unlikely to have occurred by chance. In contrast, larger P-values imply that the trend may not be statistically significant and could result from random variation.

It is important to note that, for the analysis of interannual variations in background O<sub>3</sub> concentration, only annual data from the compiled dataset were used. To ensure robustness, individual data points that deviated markedly from the overall regional trend were excluded, whereas consecutive deviations were retained to preserve temporal continuity. Here, "deviation" refers to values that differ substantially from most annual data within the aggregated regional dataset – likely reflecting spatial heterogeneity – rather than statistical outliers at the level of individual studies. In case where background O<sub>3</sub> concentrations were estimated using multiple methods within the same geographical region of a single study, the results were averaged to provide a more representative value.

#### 3 Background ozone: conceptual evolution and key definitions

Background  $O_3$  generally refers to the portion of  $O_3$  concentrations that are not influenced by direct local anthropogenic emissions, though its definition varies across studies globally. In contemporary atmospheric research, background  $O_3$  is commonly categorized into two distinct types: natural

background ozone (NBO) and regional background ozone (RBO). These two categories are crucial for understanding the sources and variations of background O<sub>3</sub> on both local and global scales. Figure 2 presents the historical evolution of background O<sub>3</sub> concepts, including their definitions and key characteristics.

Natural background ozone (NBO) is defined as tropospheric O<sub>3</sub> that exists in the complete absence of anthropogenic emissions, originating solely from natural processes (McDonald-Buller et al., 2011; Vingarzan, 2004; Wu et al., 2008). The primary sources of NBO include VOCs and NO<sub>x</sub> emitted by natural sources such as vegetation, soil, lightning, wildfires, and the oxidation of CH<sub>4</sub>, as well as O<sub>3</sub> exchange between the stratosphere and troposphere (Thompson, 2019). Historically, research into NBO originated with studies on atmospheric photochemistry. In the 1950s, investigations into photochemical smog in Los Angeles identified O<sub>3</sub> as a major component of smog, linking vehicular emissions of VOCs and NO<sub>x</sub> to its formation (Haagen-Smit, 1952). While these studies primarily focused on anthropogenic sources, they also observed detectable O<sub>3</sub> concentrations in remote regions, far from urban pollution, suggesting natural processes contributed to O<sub>3</sub> production (Galbally et al., 1986; Volz and Kley, 1988). By the late 1970s, systematic studies in the U.S. identified key natural sources of O<sub>3</sub>, such as biogenic VOCs (BVOCs), lightning, and soil-emitted NO<sub>x</sub>, leading to the formation of the NBO concept (Crutzen, 1974; Jacob et al., 1999; Liu et al., 1987). Although NBO holds significant scientific importance, its practical application as a regulatory tool remains limited, particularly in the Northern Hemisphere, where anthropogenic emissions dominate regional O<sub>3</sub> production (Berlin et al., 2013). Nonetheless, NBO is a critical reference for establishing baseline O<sub>3</sub> levels globally, facilitating the evaluation of human contribution to atmospheric O<sub>3</sub> concentration.

In the 1990s, researchers in the U.S. began to recognize the critical role of long-range transport from anthropogenic sources in regional O<sub>3</sub> levels (Fiore et al., 2002a; Jacob et al., 1999; Vingarzan, 2004). This realization was pivotal in developing the concept of United States Background Ozone (USBO), which includes O<sub>3</sub> contributions from global NBO as well as anthropogenic emissions originating outside the country, such as from neighboring regions like Canada and Mexico (Skipper et al., 2021; Thompson, 2019). Acknowledging these external sources highlighted that background O<sub>3</sub> levels could not be fully

mitigated through domestic emission reductions alone.

et al., 2024; Wang et al., 2022a).

By the early 21st century, research on background O<sub>3</sub> increasingly intersected with air quality policy development. A notable milestone was the introduction of Policy Relevant Background Ozone (PRBO) by the United States Environmental Protection Agency (EPA) in 2006 during revisions to the National Ambient Air Quality Standards (NAAQS) (U.S. EPA, 2006; Zhang et al., 2011). PRBO refers to groundlevel O<sub>3</sub> concentrations that exclude all anthropogenic emissions from North America (the U.S., Canada and Mexico) while accounting for natural sources and long-range transport from anthropogenic and natural sources outside North America (Emery et al., 2012; Nopmongcol et al., 2016). This concept aimed to help policymakers assess the effectiveness of domestic control measures in reducing O<sub>3</sub> pollution and inform the establishment of stricter O<sub>3</sub> standards. By differentiating controllable from uncontrollable O<sub>3</sub> sources, PRBO enabled a more targeted approach to air quality management, framing policy discussions around the limitations of local pollution control in addressing O<sub>3</sub> levels (Duc et al., 2013; Zhang et al., 2011). The introduction of PRBO marked a significant transition in background O<sub>3</sub> research, shifting from a predominantly scientific focus to one directly informing air quality policy and regulatory frameworks (Hosseinpour et al., 2024; U.S. EPA, 2006, 2007). Although USBO and PRBO share some common elements, their definitions differ primarily in geographic scope. PRBO focuses on transboundary contributions from regions outside North America, whereas USBO includes emissions from neighboring countries, such as Canada and Mexico, that affect the U.S. O<sub>3</sub> concentration. To address regional variations and better capture the dynamic of background O<sub>3</sub> in specific areas, advancements in atmospheric chemistry models have enabled scientists to differentiate the contributions of various sources to background O<sub>3</sub>. This led to the emergence and widespread adoption of the term Regional Background Ozone (RBO) around the 2010s (Kemball-Cook et al., 2009; Langford et al., 2009; Ou-Yang et al., 2013). RBO refers to O<sub>3</sub> concentrations within a defined region that are unaffected by direct local anthropogenic emissions. Its main sources include natural emissions (e.g., BVOCs, soil, wildfires, and lightning), the oxidation of CH4, stratospheretroposphere exchange, and long-range transport (McDonald-Buller et al., 2011; Skipper et al., 2021; Sun The distinction between NBO and RBO is crucial for understanding the complexity of background O<sub>3</sub> concentrations, as each reflects different sources and scales of influence. NBO represents a natural baseline, dominated by non-anthropogenic factors, serving as a reference point for assessing the human impact on atmospheric composition. In contrast, RBO reflects the interplay of natural and anthropogenic sources at local and global scales. Advancing our understanding of both NBO and RBO is essential for improving air quality models, refining emission control strategies, and establishing science-based standards for O<sub>3</sub> pollution reduction.

Figure 2: Historical evolution of background O<sub>3</sub> concepts: definitions (left box) and characteristics (right box).

## 4 Methods for estimating background ozone concentrations

The estimation of regional background O<sub>3</sub> is typically conducted using four primary methods: (1) in situ measurement estimation, (2) statistical analysis estimation, (3) numerical modeling estimation, and (4) integrated methods estimation. Figure 3 summarizes the advantages, limitations, and applicability of each method, providing a comparative overview of their respective strengths and weaknesses.

## 4.1 In situ measurement estimation

The in situ measurement estimation method involves the deployment of monitoring stations in remote or elevated areas, typically located far from direct pollution sources, to measure O<sub>3</sub> concentrations directly (Lam and Cheung, 2022; Wang et al., 2009b). This approach is widely recognized as one of the most direct and commonly used methods for estimating regional background O<sub>3</sub>. It is relatively straightforward to implement, requires minimal post-measurement processing, and provides continuous, high frequency data on O<sub>3</sub> variations across spatial and temporal scales. These attributes render it an invaluable tool for tracking long-term trends in background O<sub>3</sub> concentrations. However, this method has limitations, particularly concerning the spatial representativeness of the data. The limited number of monitoring stations, especially in regions with complex terrain or vast geographic areas, can result in insufficient coverage of the region's environmental conditions. Furthermore, measurements from background stations are subject to local meteorological conditions, such as temperature, humidity, and wind patterns, which can introduce uncertainties into background O<sub>3</sub> concentrations estimates (Skipper et al., 2021; Wu et al., 2017). This challenge is particularly pronounced in the Northern Hemisphere, where widespread anthropogenic emissions complicate the identification of truly "background" stations that are unaffected by human activities (Cooper et al., 2012; McDonald-Buller et al., 2011; Skipper et al., 2021; Vingarzan, 2004). Despite its limitations, the in situ measurement estimation method remains an indispensable tool for estimating background O<sub>3</sub> concentrations. For instance, Vingarzan (2004) reported that background O<sub>3</sub> concentration in the Northern Hemisphere rose from approximately 10 ppb before the Industrial Revolution to 25–40 ppb by the 2000s, corresponding to an annual growth rate of 0.5–2%. Similarly, Akimoto et al. (2015) found background O<sub>3</sub> concentrations ranging from 60 to 70 ppb in Japan's Tokyo and Fukuoka metropolitan areas between 1990 and 2008. In southern China, Wang et al. (2009b) recorded background O<sub>3</sub> levels of 30-40 ppb at the Hok Tsui station in Hong Kong from 1994 to 2018, with an average annual increment of 0.58 ppb. These studies demonstrate that, despite challenges in achieving complete representativeness, the in situ measurement estimation method provides valuable insights into

regional background O<sub>3</sub> trends and advances our understanding of the long-term impacts of both natural

and anthropogenic processes on atmospheric chemistry.

## 4.2 Statistical analysis estimation

The statistical analysis estimation method uses observed O<sub>3</sub> concentration data and applies statistical techniques to estimate regional background O<sub>3</sub> levels (Altshuller and Lefohn, 1996; Berlin et al., 2013; Steiner et al., 2010; Wang et al., 2022a). Historically, such estimations primarily relied on real-time measurements from monitoring stations. However, limitations in the spatial and temporal coverage of monitoring networks, along with their susceptibility to local environmental factors, have constrained their ability to capture the broader regional O<sub>3</sub> levels accurately. For example, monitoring stations situated in areas with complex terrain may yield skewed data due to topographical effects on air circulation patterns, which in turn significantly influence the distribution of O<sub>3</sub> concentration (Wang et al., 2022a). To overcome these challenges, researchers have increasingly adopted advanced statistical models that incorporate diverse observational data sources, enhancing the accuracy and reliability of background O<sub>3</sub> estimates (Riley et al., 2023; Rizos et al., 2022).

A notable advantage of statistical analysis estimation methods is their capability to process

A notable advantage of statistical analysis estimation methods is their capability to process extensive datasets over long temporal scales, providing a cost-effective approach to estimating regional background O<sub>3</sub> levels. These methods can leverage large-scale data networks, such as satellite observations or regional monitoring systems (Langford et al., 2009). However, the reliability of statistical models is heavily dependent on the quality and spatial representativeness of the input observational data. High quality data are essential to minimize biases, and the monitoring stations must be strategically distributed to represent the target region adequately. Additionally, rigorous data preprocessing is critical to mitigate the influence of external factors, such as extreme weather events, that may distort the background O<sub>3</sub> concentrations estimates (Berlin et al., 2013; Langford et al., 2009).

The commonly used statistical analysis methods include the following:

## 4.2.1 Principal Component Analysis

Principal Component Analysis (PCA) is a widely used multivariate statistical technique designed to extract key patterns from datasets containing multiple interrelated variables (Jolliffe, 2005). By

transforming correlated variables into a smaller set of uncorrelated principal components, PCA effectively reduces data complexity while preserving the most significant information. In the context of atmospheric pollution, PCA has proven to be particularly useful for isolating background O<sub>3</sub> by minimizing the influences from meteorological factors, such as temperature, humidity, and wind, as well as local airflows from urban and industrial sources. This makes PCA an invaluable tool for understanding regional air quality and estimating background O<sub>3</sub> levels, particularly in cases where direct measurements are confounded by local pollution or short-term meteorological variability.

## 4.2.2 K-means clustering

K-means clustering is an unsupervised, iterative machine-learning algorithm widely employed for grouping data, such as O<sub>3</sub> concentrations, meteorological parameters, and other environmental factors, based on shared characteristics (Riley et al., 2023). Clusters with minimal anthropogenic influence are often interpreted as representative of background O<sub>3</sub> concentrations. These clusters, typically defined by low pollutant levels or specific meteorological conditions, facilitate the identification of periods or locations where regional background O<sub>3</sub> can be reliably assessed (Riley et al., 2023; Zohdirad et al., 2022).

## 4.2.3 TCEQ method

The Texas Commission on Environmental Quality (TCEQ) method, based on O<sub>3</sub> monitoring data from background regions, has been widely adopted in Texas, as a reliable approach for estimating regional background O<sub>3</sub> levels (Nielsen-Gammon et al., 2005). This approach defines regional background O<sub>3</sub> as the minimum value within maximum daily 8-hour average (MDA8) O<sub>3</sub> across all monitoring stations in a given area, effectively representing the lowest O<sub>3</sub> levels unaffected by local emissions (Wu et al., 2017). By focusing on these minimum values over an extended period, the TCEQ method isolates background concentrations, which are crucial for understanding regional air quality and evaluating long-term trends in O<sub>3</sub> pollution.

## 4.2.4 O<sub>3</sub>-NO<sub>z</sub> intercept method

The O<sub>3</sub>-NO<sub>z</sub> intercept method is an approach for estimating background O<sub>3</sub> concentrations by establishing the linear relationship between O<sub>3</sub> concentrations and its precursors (Altshuller and Lefohn, 1996; Hirsch et al., 1996; Yan et al., 2021). In this approach, NO<sub>z</sub> is defined as the difference between NO<sub>y</sub> (the total reactive nitrogen species, including nitric acid and peroxy nitrates) and NO<sub>x</sub> (which comprises NO and NO<sub>2</sub>). NO<sub>z</sub> serves as an indirect indicator of background O<sub>3</sub> level, based on the assumption that it reflects the presence of O<sub>3</sub>-producing precursors in the atmosphere. Through regression analysis, O<sub>3</sub> levels are extrapolated to the intercept where NO<sub>z</sub> equals zero, representing an approximation of background O<sub>3</sub> concentrations unaffected by local emissions and photochemical influences.

However, Yan et al. (2021) noted that the method's accuracy could be compromised in areas with high rates of nitric acid (HNO<sub>3</sub>) deposition. Elevated HNO<sub>3</sub> deposition sequesters reactive nitrogen compounds at the surface, potentially masking near-surface O<sub>3</sub> levels and leading to overestimations of background O<sub>3</sub> concentrations. To address these limitations, Yan et al. (2021) proposed a modified version of the O<sub>3</sub>-NO<sub>z</sub> method, referred to as the 1-σ O<sub>3</sub>-NO<sub>z</sub> method. This refinement involved excluding regions with high HNO<sub>3</sub> deposition rates and minimizing the influence of regional emissions through improved data selection criteria.

## 4.2.5 O<sub>3</sub>-CO-HCHO response method

Cheng et al. (2018) introduced an innovative approach for estimating background O<sub>3</sub> concentrations by using carbon monoxide (CO) and formaldehyde (HCHO) as chemical indicators to trace the production and consumption of O<sub>3</sub>. This method integrates the chemical reaction dynamics between O<sub>3</sub>, CO, and HCHO, resulting in a rapid-response O<sub>3</sub> estimator. This approach was specifically designed to enhance the efficiency and accuracy of O<sub>3</sub> estimation by leveraging the dynamic chemical processes that influence O<sub>3</sub> levels. Building upon this foundation, Yan et al. (2021) proposed the O<sub>3</sub>-CO-HCHO approach, which refines the original concept by eliminating the influence of both anthropogenic and natural emissions of O<sub>3</sub> precursors, enabling a more accurate estimation of background O<sub>3</sub> concentrations.

The O<sub>3</sub>-CO-HCHO method is particularly advantageous due to its applicability to both observational data and model outputs, offering robust results for regions with high isoprene emissions.

The method is governed by the following key equations:

$$0_{3} = k_{1}(CO_{total} - CO_{back}) - (k_{1}k_{2} - k_{3})(HCHO_{total} - HCHO_{back}) + O_{3back},$$
 (2)

$$O_{3back} = O_3 - k_1(CO_{total} - CO_{back}) + (k_1k_2 - k_3)(HCHO_{total} - HCHO_{back}),$$
(3)

- Here,  $k_1 = \frac{\Delta O_3}{\Delta CO_{anthro}}$ ,  $k_2 = \frac{\Delta CO_{bio}}{\Delta HCHO_{bio}}$ ,  $k_3 = \frac{\Delta O_3}{\Delta HCHO_{bio}}$ . The terms "anthro", "bio", "total", and
- "back" refer to anthropogenic sources, biogenic sources, total sources, and background sources,
- respectively.

361

370

#### 4.2.6 Percentile method

- The percentile method is a widely adopted statistical approach for estimating regional background O<sub>3</sub>
- concentrations, offering a straightforward and practical alternative to complex modeling techniques
- (Berlin et al., 2013; Jenkin, 2008). This method involves analyzing O<sub>3</sub> concentration data over a specific
- time period and selecting a particular percentile to represent the background O<sub>3</sub> levels. The selected
- percentile is assumed to reflect minimal O<sub>3</sub> concentrations that are largely unaffected by local pollution
- sources, thereby serving as a proxy for regional background O<sub>3</sub> concentrations.

## 4.2.7 Temperature-ozone relationship method

- The temperature-ozone relationship method estimates background O<sub>3</sub> contributions by analyzing the
- correlation between O<sub>3</sub> concentrations and temperature (Mahmud et al., 2008). Generally, O<sub>3</sub>
- concentrations increase with rising temperatures, as elevated temperatures enhance the photochemical
- reactions that produce O<sub>3</sub>. However, within a specific temperature range, O<sub>3</sub> concentrations tend to
- stabilize due to the equilibrium between O<sub>3</sub> production and destruction processes. These stabilized O<sub>3</sub>
- levels, typically observed during periods of relatively stable meteorological conditions, are often
- regarded as indicative of regional background O<sub>3</sub> concentrations, reflecting natural influence rather than
- anthropogenic emissions (Mahmud et al., 2008; Sillman and Samson, 1995; Steiner et al., 2010).

## 4.2.8 Nocturnal ozone concentration method

- The nocturnal O<sub>3</sub> concentration method leverages the relatively stable O<sub>3</sub> levels observed during
- nighttime, when photochemical reactions driven by sunlight are absent, making it a valuable approach

for estimating regional background O<sub>3</sub> levels (Chan et al., 2003). At night, O<sub>3</sub> levels generally remain constant or exhibit minimal fluctuations, as they are primarily governed by the equilibrium between O<sub>3</sub> production and destruction through reactions with NO<sub>x</sub> and other atmospheric components. However, this method is not without its challenges. A key limitation arises from the titration reaction between O<sub>3</sub> and NO, which produces NO<sub>2</sub> and depletes ambient O<sub>3</sub> levels. This phenomenon, known as O<sub>3</sub> titration, can result in underestimation of true background O<sub>3</sub> concentrations, particularly in areas with elevated NO emissions (Akimoto et al., 2015; Itano et al., 2007; Shin et al., 2012).

To mitigate the impact of  $O_3$  titration, researchers have introduced adjustments to nocturnal  $O_3$  estimates by incorporating a "total  $O_3$ " concentration, denoted as  $O_{3 \text{ total}}$ , which serves as a proxy for background  $O_3$  levels. The "total  $O_3$ " is calculated using the following equations:

$$\left[0_{3_{\text{total}}}\right] = \left[0_{3}\right] + \left[NO_{2}\right] - \alpha \times \left[NO_{x}\right],\tag{4}$$

Here,  $[O_3]$ ,  $[NO_2]$ , and  $[NO_x]$  (=  $[NO] + [NO_2]$ ) represent the mixing ratios of  $O_3$ ,  $NO_2$ , and  $NO_x$ , respectively. The parameter  $\alpha$  accounts for the fraction of  $NO_2$  in  $NO_x$  from primary emissions, with a typical value of  $\alpha = 0.1$  used in most studies (Akimoto et al., 2015; Itano et al., 2007; Shin et al., 2012). However, Wang et al. (2009b) suggested a lower value of  $\alpha = 0.041$ , introducing variability in the estimated  $[O_{3total}]$ . This adjustment helps to compensate for the effects of NO titration, yielding a more accurate representation of regional background  $O_3$  levels.

Statistical analysis methods have been widely used to estimate regional background O<sub>3</sub> concentrations. For example, Langford et al. (2009) applied PCA to analyze regional background O<sub>3</sub> concentrations in Texas from August to October 2006. Their analysis revealed that the first principal component accounted for approximately 84% of the variance in the O<sub>3</sub> data, strongly indicating its relevance as a proxy for background O<sub>3</sub> levels. Riley et al. (2023) applied K-means clustering to estimate background O<sub>3</sub> concentrations in eastern Australia from 2017 to 2022. Their analysis revealed an average background O<sub>3</sub> concentration of 28.5 ppb, with a decadal increase of 1.8 ppb, reflecting the global trend of rising background O<sub>3</sub> levels. Berlin et al. (2013) and Langford et al. (2009) used TCEQ method to estimate background O<sub>3</sub> concentrations during high-O<sub>3</sub> periods (May–October) in Texas between 2000 and 2012. Their estimates ranged from 25 to 45 ppb and 40 to 80 ppb, respectively. Akimoto et al. (2015)

proposed using the 2<sup>nd</sup> percentile of MDA8 O<sub>3</sub> concentrations as a suitable measure of background O<sub>3</sub> levels in Japan, capturing low concentrations unaffected by local anthropogenic emissions during high-O<sub>3</sub> episodes. Chen et al. (2022) used temperature-ozone relationship method to assess background O<sub>3</sub> levels of the region-specific in China from 2013 to 2019, reporting concentrations of 35–40 ppb during clean seasons and 50–55 ppb during O<sub>3</sub>-polluted seasons.

## 4.3 Numerical modeling estimation

The numerical modeling estimation method, which uses atmospheric chemistry and transport models such as GEOS-Chem, WRF-Chem, and CMAQ, is widely employed to simulate the formation, transportation, and variability of regional background O<sub>3</sub> concentrations. These models offer several distinct advantages by incorporating a comprehensive array of atmospheric processes, including photochemical reactions, vertical mixing, advection, and the transport of pollutants across various spatial and temporal scales. By accounting for the intricate interactions among emissions, meteorological conditions, and atmospheric chemistry, numerical models provide a more robust and accurate representation of regional background O<sub>3</sub> levels compared to in situ measurement estimation or statistical analysis estimation methods alone. Additionally, numerical models can be customized to align with specific research objectives through adjustments to chemical mechanisms and parameterization schemes, rendering them adaptable to diverse regions and temporal scales.

A notable strength of numerical models lies in their ability to differentiate the contributions of various emission sources to regional O<sub>3</sub> concentrations (Jaffe et al., 2018; Thompson, 2019; Zhang et al., 2011). This capability sets them apart from in situ measurement estimation and statistical analysis estimation approaches, which typically lack the granularity to isolate the relative contributions of natural versus anthropogenic emissions. However, numerical modeling estimation also presents significant challenges. These models are computationally intensive, requiring substantial resources, especially when simulating extensive domains or prolonged time periods. Moreover, their accuracy depends heavily on the quality of input data, such as emission inventories, meteorological conditions, and assumptions regarding physical and chemical processes, which can introduce uncertainties in estimated O<sub>3</sub> concentrations (Dolwick et al., 2015; Guo et al., 2018; Hogrefe et al., 2018; Jaffe et al., 2018).

Numerical models typically estimate regional background O<sub>3</sub> concentrations using two primary approaches: the emission scenario method and the tracer method (Fiore et al., 2002a). The emission scenario method employs three-dimensional air quality models, such as GEOS-Chem, MOZART, WRF-Chem, and CMAQ, to simulate background O<sub>3</sub> levels by conducting perturbation experiments where local anthropogenic emissions are reduced or set to predefined values. This approach enables the isolation of local emissions' contributions to regional background O<sub>3</sub> levels (Zhang et al., 2011; Li et al., 2018; Lu et al., 2019a; Pfister et al., 2013). In contrast, the tracer method uses chemical tracers to track the transport and transformation of emissions, offering an alternative approach to estimating background O<sub>3</sub> concentrations. Models such as CMAQ-ISAM and CAMx-OSAT, developed by the U.S. Environmental Protection Agency (EPA), incorporate tracer methods to estimate regional background O<sub>3</sub> concentrations (Lefohn et al., 2014; Li et al., 2012; Reid et al., 2008). Although both methods have their strengths, studies have highlighted discrepancies in O<sub>3</sub> estimates depending on the approach employed (Jaffe et al., 2018; Skipper et al., 2021). For example, Emery et al. (2012) found that the CAMx model generally produced higher background O<sub>3</sub> concentrations in the U.S. compared to GEOS-Chem, with CAMx showing a higher correlation with observational data, especially at remote stations and during high-O<sub>3</sub> episodes. Conversely, GEOS-Chem demonstrated greater accuracy in capturing seasonal mean O<sub>3</sub> concentrations in rural areas. Similarly, Dolwick et al. (2015) compared the tracer and emission scenario methods using CAMx and CMAQ models. Their analysis revealed consistent estimates of background O<sub>3</sub> concentrations in suburban U.S. areas across both methods. However, in urban areas, the tracer method yielded lower background O<sub>3</sub> estimates than the emission scenario method, indicating a substantial influence of local emissions on O<sub>3</sub> concentrations in densely populated regions. Equally, Fiore et al. (2014) reported differences in background O<sub>3</sub> concentrations between GEOS-Chem and GFDL-AM3 models, with variations ranging from 1 to 10 ppb depending on region, season, and altitude. Numerical modeling estimation has been extensively applied to estimate global and regional background O<sub>3</sub> concentrations. For example, using the global model GEOS-Chem, Emery et al. (2012)

and Zhang et al. (2011) estimated average background O<sub>3</sub> concentration in the U.S. from March to August

2006, ranging from 20 to 45 ppb, with  $27 \pm 8$  ppb in low-altitude areas and  $40 \pm 7$  ppb in high-altitude areas. Guo et al. (2018) reported annual variation of up to 15 ppb in regional background  $O_3$  concentration in the U.S. between June and August from 2004 to 2012. Meanwhile, regional models such as CAMx and CMAQ yielded background  $O_3$  estimates of 25 to 50 ppb in the U.S. between March and August 2006 (Emery et al., 2012). In China, Sahu et al. (2021) found background  $O_3$  concentrations exceeded 22 ppb in 2015.

#### 4.4 Integrated methods estimation

The three methods discussed above each possess distinct advantages and limitations, contributing to uncertainties in estimating regional background O<sub>3</sub> concentration. Given these challenges, researchers have increasingly turned to integrated methods to improve the accuracy and reliability of these estimations.

For instance, Dolwick et al. (2015) improved model-based estimates of background O<sub>3</sub> by comparing observed and simulated O<sub>3</sub> concentrations. Their analysis of rural areas in the western U.S. during April to October 2007 reported background O<sub>3</sub> concentrations ranging from 40 to 45 ppb, with the lowest concentrations observed along the Pacific coast, ranging from 25 to 35 ppb.

Similarly, Sun et al. (2024) refined estimates by treating model biases as spatial functions, optimizing regional background O<sub>3</sub> estimations. Based on this methodology, Skipper et al. (2021) extended the methodology by incorporating both spatial and temporal functions to account for variations driven by regional background O<sub>3</sub> and anthropogenic emissions. This revised approach estimated an average background O<sub>3</sub> concentration of approximately 33 ppb for the U.S. in 2017, with peak values around 38 ppb. Notably, this adjustment improved the consistency of estimates by 28% compared to the unadjusted model, demonstrating the utility of integrated methods in refining atmospheric models.

The rapid advancement of machine learning (ML) techniques has further facilitated the integration of these technologies with traditional methods for estimating regional background O<sub>3</sub> concentrations. For example, Hosseinpour et al. (2024) developed a multivariate linear regression (MVLR) model and a random forest (RF) based ML algorithm to adjust model-derived background O<sub>3</sub> concentrations. While the MVLR model follows an adjustment method akin to that of Skipper et al. (2021), the RF-ML

algorithm employs the Shapley Additive Explanations (SHAP) method to evaluate the relative importance of each input variable. The RF-ML model, trained using k-fold cross-validation, demonstrated superior predictive accuracy. Hosseinpour et al. (2024) showed that the RF-ML algorithm produced results most consistent with those from the in situ measurement estimation method, outperforming those from the original CAMx model, MVLR adjustments, and two other ML algorithms. Utilizing this methodology, they estimated background O<sub>3</sub> concentrations in 13 urban areas of the U.S. during April–June and July–September 2016 to range from 31–46 ppb and 27–45 ppb, respectively. This finding underscores the potential of ML algorithms to enhance model-based background O<sub>3</sub> estimates by capturing nonlinear relationships and complex variable interactions (Breiman, 2001; Kashinath et al., 2021).

Overall, integrated methods, particularly those integrated with machine learning techniques, represent a significant advancement in estimating regional background O<sub>3</sub> concentrations. These approaches not only improve the accuracy and robustness of estimates but also provide valuable insights into the complex dynamics of O<sub>3</sub> formation and transport. By combining observational data, statistical adjustments, and advanced modeling techniques, researchers can achieve a more comprehensive understanding of regional O<sub>3</sub> levels and their temporal variations.

Figure 3: Summary of the advantages, limitations, and applicability of different estimation methods for background O<sub>3</sub>.

## 5 Comprehensive assessments of background ozone in China: patterns, trends, sources, and global comparisons

5.1 Regional patterns of background ozone in China Figure 4 presents the average background O<sub>3</sub> concentrations across various regions in China. On a national scale, the average background  $O_3$  concentration is  $41.4 \pm 12.2$  ppb, accounting for 79% of the MDA8 O<sub>3</sub> concentration. Notable regional variability in background O<sub>3</sub> concentrations is observed, highlighting the differential impacts of local meteorological conditions, pollutant emissions, and geographic characteristics. Among the regions, Northwest China (NWC) stands out with the highest background O<sub>3</sub> concentrations, reaching  $48.2 \pm 8.3$  ppb, which accounts for 96% of the MDA8  $O_3$  concentration. This elevated concentration is attributed to several interrelated factors. First, strong solar radiation and arid atmospheric conditions enhance photochemical reactions, accelerating O<sub>3</sub> formation. He et al. (2021) demonstrated that abundant sunshine and dry conditions significantly increase O<sub>3</sub> production due to the intensified photolysis of precursor compounds. Furthermore, the high altitude and unique surface characteristics of Northwest China (NWC) promote strong daytime atmospheric convection, facilitating the downward transport of O<sub>3</sub> from the upper atmosphere to the surface levels (Ding and Wang, 2006; Liu et al., 2019; Ma et al., 2005; Nie et al., 2004). Additionally, the relatively low anthropogenic emissions result in fewer precursors like NOx and VOCs, thereby minimizing rapid fluctuations in O3 levels. The weaker nocturnal O<sub>3</sub> depletion, caused by limited O<sub>3</sub> scavenging from sparse emissions and lower nighttime temperatures, further amplifies baseline O<sub>3</sub> concentrations (Nie et al., 2004; Qin et al., 2023; Xu et al., 2020). The urban clusters of North China (NC) and East China (EC), along with Southwest China (SWC), also exhibit higher background O<sub>3</sub> concentrations, averaging  $40.3 \pm 14.9$  ppb,  $39.0 \pm 13.4$  ppb, and 38.4 $\pm$  10.4 ppb, respectively. These concentrations account for 75%, 67%, and 83% of the MDA8  $O_3$ concentration in each respective region. East China (EC) and North China (NC) are heavily influenced by high industrial and vehicular emissions, which release significant quantities of NO<sub>x</sub> and VOCs. The

precursors undergo photochemical reactions under intense sunlight and elevated summer temperatures,

resulting in higher O<sub>3</sub> levels. Moreover, the East Asian Summer Monsoon (EASM) facilitates the transport of O<sub>3</sub> and its precursors from low-latitude regions, such as South China (SC), to higher latitudes, exacerbating O<sub>3</sub> pollution during the monsoon season (Gao et al., 2005; Liu et al., 2019, 2021; Sun et al., 2016; Xu et al., 2011, 2020). In contrast, in Southwest China (SWC), regional pollutant transport plays a significant role. During spring, prevailing winds carry pollutants such as NO<sub>x</sub> and VOCs from Southeast Asia, intensifying local O<sub>3</sub> levels (Ye et al., 2024). Summer conditions – characterized by high humidity, elevated temperatures, and intense solar radiation – further amplify photochemical O<sub>3</sub> formation (Chen, 2020). The region's complex topography, including mountainous areas and plateaus, also contributes to localized O<sub>3</sub> accumulation. For instance, the Sichuan Basin, with its basin-like terrain, impedes air mass dispersion, leading to pollutants entrapment and prolonged O<sub>3</sub> buildup (Hu et al., 2019). The background O<sub>3</sub> concentrations in South China (SC), Central China (CC), and Northeast China (NEC) are relatively low compared to other regions of China, with values of  $37.0 \pm 8.9$  ppb,  $35.1 \pm 12.6$ ppb, and  $33.1 \pm 5.7$  ppb, respectively. These concentrations account for 74%, 60%, and 68% of the MDA8 O<sub>3</sub> in each corresponding region. In South China (SC), the relatively low background O<sub>3</sub> concentrations can be primarily attributed to the frequent rainfall and high humidity, which facilitate the removal of O<sub>3</sub> precursors such as NO<sub>x</sub> and VOCs, thereby suppressing O<sub>3</sub> formation (He et al., 2021). Although BVOCs emissions are relatively high in this region due to abundant vegetation and elevated temperatures, their impact on O<sub>3</sub> formation is less pronounced compared to regions like North China (NC). This is because anthropogenic emissions, such as vehicular exhaust and industrial discharges, typically amplify the contribution of BVOCs to O<sub>3</sub> formation. In the absence of significant anthropogenic pollution, the role of BVOCs in O<sub>3</sub> formation remains relatively limited (Ye et al., 2024). In Central China (CC), the lower background O3 concentrations are linked to the region's inland locations, which reduce its exposure to oceanic influences and transboundary pollutant transport. The absence of strong maritime airflow limits the import of O<sub>3</sub> precursors, while frequent rainfall during the warmer months helps remove these precursors from the atmosphere, further suppressing O<sub>3</sub> formation (Sahu et al., 2021; Ma et al., 2024). Anthropogenic emissions, primarily from vehicular exhaust, industrial

discharges, and solvent usage, constitute the dominant sources of O<sub>3</sub> in this region (Zeng et al., 2018).

Consequently, the relative contribution of background O<sub>3</sub> is lower, as anthropogenic emissions play a more substantial role in O<sub>3</sub> formation. In Northeast China (NEC), the lower background O<sub>3</sub> concentration can be attributed to a prolonged period of low temperature, which significantly reduces the rate of photochemical reaction. Additionally, the region experiences strong summer air convection and substantial precipitation, both of which further inhibit O<sub>3</sub> generation (Chen, 2024; Xu et al., 2020).

Figure 4: Spatial distribution of background O<sub>3</sub> concentrations (1994–2020) across various regions of China. The locations of 33 background monitoring stations are indicated with red dots. The seven regions include Northeast China (NEC), North China (NC), East China (EC), Central China (CC), Northwest China (NWC), Southwest China (SWC), and South China (SC).

## 5.2 Comparative evaluation of background ozone concentration estimates using diverse methods

Figure 5 presents a comparative assessment of background  $O_3$  concentrations estimates in China from four common approaches: in situ measurement, statistical analysis, numerical modeling, and integrated methods. Among these, in situ measurement estimation method remains the most widely applied, supported by extensive datasets from 33 background monitoring sites (n = 678; Fig. 4, Table S3). By contrast, integrated methods have only recently emerged and have been applied in a limited number of case (n=8), reflecting their greater methodological complexity and reliance on comprehensive data

## integration.

National mean background  $O_3$  concentrations estimated by different methods are broadly comparable but show notable differences. In situ measurement estimation method (41.7  $\pm$  12.3 ppb) and statistical analysis estimation method (39.9  $\pm$  11.3 ppb) yield the highest values, followed by numerical modeling estimation method (37.4  $\pm$  11.9 ppb). Integrated methods yield the lowest values of 34.5  $\pm$  1.6 ppb, approximately 6 ppb lower than those from in situ measurement estimation and statistical analysis estimation method.

Despite similar mean values, the variability across methods is substantial. In situ measurement estimation reveals a particularly wide variability, with estimated background O<sub>3</sub> concentrations ranging from approximately 14 ppb to as high as 85 ppb. This broad range reflects the substantial influence of localized factors, such as topography, climatic conditions, and anthropogenic emissions, on observational data. In comparison, statistical analysis estimation and numerical modeling estimation methods yield relatively consistent results, although the difference between the maximum and minimum estimated background O<sub>3</sub> concentration for both methods reaches 60 ppb. Notably, more than 80% of the estimated background O<sub>3</sub> concentrations fall within the range of 25–53 ppb, suggesting a reasonable degree of agreement between the two methods. The consistency is likely attributable to the reliance on long-term data trends and calibrated algorithms, which effectively reduce the impact of extreme values while capturing broader patterns in O<sub>3</sub> behavior.

In contrast, the integrated methods – combining in situ observation, statistical analysis, and numerical results – yield the narrowest range (32–37 ppb), with the value of  $34.5 \pm 1.6$  ppb. This narrow range reflects their strength in reconciling model consistency with real-world variability, rather than oversimplification. By harmonizing data sources, integrated methods reduce methodological noise and yield more robust, policy-relevant estimates. The limited number of applications, however, may also contribute to the observed low variability. Although studies in China remain scarce, international applications underscore their potential. For instance, Skipper et al. (2021) showed that incorporating spatial and temporal bias corrections improved the consistency of model-derived background  $O_3$  estimates by 28% relative to unadjusted models. Similarly, Hosseinpour et al. (2024) demonstrated that

a random forest machine learning (RF-ML) algorithm integrating multiple data sources with nonlinear feature analysis produced background O<sub>3</sub> estimates most consistent with in situ observations for correcting air quality model simulations and outperformed the original CAMx model, multivariate regression, and other ML algorithms. Collectively, these studies highlight the value of integrated methods in producing consistent estimates, particularly for regulatory applications and long-term trend assessments. Nevertheless, further validation is needed to determine whether the observed low variability reflects true methodological robustness or limited sampling. Importantly, no single method is definitive. Each carries inherent assumptions. Integrated methods therefore provide a complementary framework that balances empirical realism with generalizability.

Method-dependent discrepancies underscore the complexity of estimating background O<sub>3</sub>. Variability arises from differences in input data, model assumptions, and the parameterization of physical and chemical processes (Jaffe et al., 2018; Skipper et al., 2021; Wang et al., 2022a; Yan et al., 2021). For instance, in situ measurement estimation method is directly influenced by local meteorological and emission conditions, whereas the numerical modeling estimation method is subject to uncertainties in simulating processes such as natural emissions, transboundary transport, and photochemical reactions. Ideally, direct comparison of background O<sub>3</sub> estimates derived from multiple methods at the same location would clarify their relative strengths and limitations. However, such comparison was not feasible here due to methodological and data constraints. First, the dataset used in this study is limited to China, where only a subset of the methods described in Sect. 4.2 has been applied, each requiring specific datasets and exhibiting region-dependent applicability. Second, background O<sub>3</sub> exhibits pronounced spatial and temporal variability, while existing studies often target different subregions and time periods, making consistent co-located comparisons impractical. Despite these challenges, several studies have conducted preliminary intercomparisons within the same region. In Shandong, Wang et al. (2022a) reported that PCA (using ambient O<sub>3</sub> alone) yielded background O<sub>3</sub> about 20 ppb higher than the TCEQ approach, with seasonal patterns more consistent with background-site observations. The TCEQ method tended to underestimate background O<sub>3</sub> because minimum MDA8 O<sub>3</sub> values were often influenced by residual urban emissions. In the inland southeastern U.S., Yan et al. (2021) found the O<sub>3</sub>-CO-HCHO method yielded the lowest estimates (10-15 ppb), the  $1-\sigma$  O<sub>3</sub>-NO<sub>z</sub> method intermediated values (15-25 ppb), and the 5<sup>th</sup> percentile method the highest (20-30 ppb), likely due to anthropogenic influences in urban downwind regions. Likewise, Chen et al. (2022) revealed that the nocturnal O<sub>3</sub> method underestimated background O<sub>3</sub> by up to 30% compared with the temperature-ozone relationship method during polluted seasons in China.

Collectively, these studies demonstrate that methodological choices alone can lead to discrepancies of 10–20 ppb in background O<sub>3</sub> estimates within the same region. Careful interpretation therefore requires explicit attention to methodological assumptions, data representativeness, and sensitivity to emission influences. Moving forward, the development of harmonized datasets would enable the consistent application of multiple methods at the same regions and time periods, providing more robust intercomparisons and clearer insights into the strengths and limitations of each approach.

Figure 5: Estimated regional average background O<sub>3</sub> concentrations in China from 1994 to 2020 based on multiple methods. All data sources are compiled and summarized in Table S1. The values of "n =" below each box indicate the number of individual data records used in the analysis for each method category.

## 5.3 Long-term trends and interannual variability of background ozone in China

Due to the absence of long-term background O<sub>3</sub> records for other regions, Figure 6 focuses on the annual

variation trends of background O<sub>3</sub> concentrations in four regions of China – South China (SC), Northwest China (NWC), North China (NC), and East China (EC) – during the period from 1994 to 2020. Overall, background O<sub>3</sub> concentrations have exhibited an upward trend across these regions, though the magnitude and significance of the trends vary regionally.

South China (SC) exhibited the most pronounced increase, with an average growth rate of 0.36 ppb yr<sup>-1</sup> (r<sup>2</sup>=0.38, p<0.01), as shown in Fig. 6(a). Although a modest decline has occurred since 2014, the long-term trend remains upward. This increase is likely driven by the regional transport of O<sub>3</sub> and its precursors. Previous studies suggest that rising background O<sub>3</sub> levels in Hong Kong are largely attributable to enhanced upwind emissions from mainland China and cross-boundary transport of precursors from Southeast Asia, particularly the Indochinese Peninsula (Wang et al., 2009b; Lee et al., 2014). Yang et al. (2019) further demonstrated that precursor emissions outside the Pearl River Delta region significantly contribute to local O<sub>3</sub> levels, with this influence intensifying in recent years.

Both Northwest China (NWC) and North China (NC) also exhibit substantial increases, with a growth rate of 0.32 ppb yr<sup>-1</sup> (r<sup>2</sup>=0.68, p<0.01) and 0.31 ppb yr<sup>-1</sup> (r<sup>2</sup>=0.34, p<0.05), respectively, as shown in Fig. 6(b) and Fig. 6(c). These trends are likely linked to enhanced stratosphere–troposphere exchange (STE) and the long-range transport of  $O_3$  precursors, as previously reported (Xu et al., 2018; Zhang et al., 2020). Large-scale circulation shifts and more frequent STE events have further amplified background  $O_3$  levels in these inland regions (Xu et al., 2016, 2020; Xue et al., 2011). Notably, for North China (NC), two separate trend lines are presented in Figure 6(c), reflecting methodological differences among studies: Ma et al. (2016) provided a long-term record using MDA8  $O_3$  concentrations filtered from in situ observations, while most other studies used hourly averages over shorter or discontinuous periods. Since MDA8  $O_3$ -based estimates are inherently higher than hourly means, aggregating them would bias trend interpretation. Therefore, separate presentation ensures consistency. Furthermore, MDA8  $O_3$  records are scarce elsewhere (typically fewer than four data points), precluding dual-trend comparison. The results of Ma et al. (2016) also support the intensification of background  $O_3$  pollution in North China (NC), reporting a much steeper growth rate of 1.35 ppb yr<sup>-1</sup> (r<sup>2</sup> = 0.80, p < 0.01) based on observations at the Shangdianzi regional background station. This suggests that both regional

emissions and conducive meteorological conditions have played synergistic roles in driving the escalation of background O<sub>3</sub> levels in North China (NC).

In contrast, East China (EC) exhibited the slowest increase in background O<sub>3</sub> concentration, with an average growth rate of 0.27 ppb yr<sup>-1</sup> that is not statistically significant (p > 0.05) (Fig. 6(d)). Several factors likely explain this muted growth (Liu et al., 2021; Xu et al., 2020; Zhang et al., 2020). First, East China (EC) was among the earliest regions in China to adopt coordinated NO<sub>x</sub> and VOCs controls, notably under the "Atmospheric Ten Measures" (2013) and the "Blue Sky Protection Campaign" (2018), which likely curbed precursors increases. Second, the region's dense urbanization and heavy industrialization complicates separation of background O<sub>3</sub> from local anthropogenic signals, potentially leading to underestimation of long-term growth. Third, meteorological conditions – higher relative humidity, more frequent precipitation, and weaker solar radiation – tend to suppress photochemical O<sub>3</sub> formation relative to drier, high-insolation regions such as North China (NC) and Northwest China (NWC). Taken together, these factors may explain why East China (EC) appears to be approaching a plateau phase in background O<sub>3</sub> levels, in contrast to the stronger upward trends observed in other regions.

Figure 6: Annual trend of background O<sub>3</sub> concentrations in the SC regions (1995–2020), NWC (1994–2019), NC (2004–2020) and EC (2004–2020), estimated using multiple independent studies (detailed in Table S1). Dashed lines indicate linear regression based on available annual data points.

#### 5.4 Seasonal variation of background ozone in China

Figure 7 illustrates the seasonal variations in mean background O<sub>3</sub> concentrations across China and its 689 seven subregions during 1994–2020. Nationally, background O<sub>3</sub> exhibits pronounced seasonality, with 690 comparable peaks in spring  $(47.2 \pm 10.6 \text{ ppb})$  and summer  $(47.3 \pm 15.4 \text{ ppb})$ , and a pronounced minimum 691 in winter  $(33.2 \pm 9.8 \text{ ppb})$ . 692 Regional patterns reveal clear differences in seasonal maxima. In Southwest China (SWC) and 693 Northeast China (NEC), peaks occurred in spring (52.1  $\pm$  9.9 ppb and 38.8  $\pm$  4.4 ppb, respectively), 694 largely driven by stratosphere-troposphere exchange (STE) and enhanced downward transport over 695 elevated terrain, and also influenced by prevailing winds that transport NO<sub>x</sub> and VOCs from Southeast 696 Asia and other regions into these areas (Liu et al., 2019; Lu et al., 2019a; Xu et al., 2018; Wang et al. 697 2011; Ye et al., 2024). In contrast, North China (NC), Northwest China (NWC), and East China (EC) 698 recorded summer maxima ( $56.8 \pm 10.8$ ,  $55.0 \pm 8.5$ , and  $48.3 \pm 16.9$  ppb, respectively), consistent with 699 the influence of the East Asian Summer Monsoon (EASM), which enhances precursor inflow and 700 stimulates photochemical O<sub>3</sub> formation under high temperatures and intense solar radiation (Gao et al., 701 2005; Liu et al., 2019, 2021; He et al., 2021). South China (SC) and Central China (CC) reached their 702 highest levels in autumn ( $46.9 \pm 10.4$  and  $43.0 \pm 14.2$  ppb, respectively), likely reflecting inland pollutant 703 transport by northeasterly winds combined with favorable sunlight conditions (Xie et al., 2022; Shen et 704 al., 2019; Luo et al., 2019). 705 Seasonal minima also varied by region. Winter lows were observed in Northeast China (NEC, 24.5 706  $\pm$  3.6 ppb), North China (NC, 24.9  $\pm$  5.2 ppb), and East China (EC, 25.2  $\pm$  8.1 ppb), reflecting weak 707 photochemistry under low temperatures and reduced solar radiation. In contrast, South China (SC, 24.8 708  $\pm$  5.0 ppb) and Central China (CC, 28.7  $\pm$  10.0 ppb) exhibited summer minima, attributable to frequent 709 precipitation and high humidity suppressing O<sub>3</sub> production. Southwest China (SWC) maintained 710 persistently low levels in both summer ( $31.0 \pm 8.2$  ppb) and autumn ( $31.0 \pm 4.6$  ppb), whereas Northwest 711 China (NWC) showed relatively lower concentrations in autumn ( $41.8 \pm 8.9$  ppb) and winter ( $41.9 \pm 5.1$ 712 ppb). 713

In summary, the seasonal cycle of background O<sub>3</sub> in China is shaped by the interplay of regional

meteorology and precursor emissions, while vertical exchange and interregional transport further modulate seasonal peaks and troughs across regions.

Figure 7: Seasonal variations in mean background  $O_3$  concentrations across seven regions of China during 1994–2020. All data sources are compiled and summarized in Table S1. The values of "n =" indicate the number of individual data records or assembly estimates used in the analysis for each region and season.

## 5.5 Source attribution and analysis of background ozone in China

The analysis above reveals that the spatiotemporal variations of background  $O_3$  are influenced by the synergistic effects of multiple factors, including regional natural source emissions, cross-regional transport, stratosphere–troposphere exchange, and local atmospheric pollutant reduction measures. These factors interact in complex and dynamic ways, resulting in significant regional and seasonal variations in background  $O_3$  levels.

Natural source emissions are a key driver of background O<sub>3</sub> levels in China, with studies consistently highlighting their substantial contribution. For example, Wang et al. (2011) and Lu et al. (2019a), using the numerical model GEOS-Chem, estimated that over 70% of regional background O<sub>3</sub> concentrations in China originate from natural emissions, including BVOCs, soil NO<sub>x</sub>, and CH<sub>4</sub> emissions and others. Among these, BVOCs exert a particularly significant impact on O<sub>3</sub> formation. Lu

et al. (2019a) demonstrated that during the peak summer months of July and August in 2016–2017, BVOCs emissions contributed over 15 ppb to the background O<sub>3</sub> in central and eastern China. Similarly, Chen et al. (2022) emphasized that during O<sub>3</sub> pollution seasons, BVOCs emissions dominate the increase in background O<sub>3</sub>, contributing 8–16 ppb compared to non-pollution seasons. These findings underscore the importance of incorporating the variability of natural emissions into modeling and policy frameworks, particularly in light of future climate change that may exacerbate BVOCs emissions.

Long-range transport plays an equally significant role in shaping background O<sub>3</sub> concentration across China. Several studies have shown that the influx of O<sub>3</sub> and its precursors from other regions, including Southeast Asia, Europe, North America, India, and the Middle East, can elevate background O<sub>3</sub> concentration in China by 2–15 ppb (Han et al., 2019; Wang et al., 2011; Wang et al., 2022b; Li et al., 2014; Ni et al., 2018). This influence is particularly pronounced during specific seasons when atmospheric circulation facilitates the transboundary transport of atmospheric pollutants (Colombi et al., 2023; Ma et al., 2025; Ni et al., 2018; Sahu et al., 2021; Ye et al., 2024). Regional transport also significantly influences the background O<sub>3</sub> levels in urbanized and densely populated areas. For instance, Su et al. (2013) showed that air masses originating from high altitudes, the Yangtze River Delta region, and the Pearl River Delta regions could cause spikes at the Mount Wuyi background station, with concentration reaching 62–73 ppb, far exceeding the station's annual average of 41 ± 15.9 ppb. Wang et al. (2022b) also found that emissions outside the Yangtze River Delta regions contributed as much as 63% to O<sub>3</sub> pollution within the region. Similarly, Wang et al. (2009b) measured that air masses from eastern China had an average O<sub>3</sub> concentration of 48 ppb at a background station in Hong Kong, highlighting the significant impact of inter-regional transport on coastal regions.

Stratosphere–troposphere exchange (STE) is a critical vertical transport process contributing to background O<sub>3</sub> levels, particularly in high-altitude and northern regions of China. This process is most active during spring, when stratospheric O<sub>3</sub> is transported downward into the troposphere (Ding and Wang, 2006; Lu et al., 2019a; Ma et al., 2025; Xu et al., 2018). Wang et al. (2011) estimated that STE contributes approximately 7 ppb to background O<sub>3</sub> concentrations in northern China during the spring season. Luo et al. (2024) further revealed that STE contributed an average of 9.6 ppb to surface O<sub>3</sub> over

the North China Plain during 19–20 May 2019. Observations at the Mt. Waliguan Station on the Tibetan Plateau further support the importance of STE; Xu et al. (2018) reported that STE contributes 8–12 ppb to background O<sub>3</sub> concentrations during spring. Lu et al. (2019a) found that STE processes contribute as much as 20 ppb to background O<sub>3</sub> concentration in western China during March and April, with an average contribution of 1.8–8.7 ppb across China from March to October. In lower-latitude regions such as the Pearl River Delta, Shen et al. (2019) demonstrated that vertical transport processes, including STE, predominantly influence background O<sub>3</sub> levels during spring and autumn. These findings underscore the critical role of altitude and latitude in modulating the magnitude of STE contributions.

#### 5.6 Comparative analysis of background ozone levels: insights from China and global perspectives

Figure 8 presents a comparative analysis of background  $O_3$  concentrations in China and several other global regions, with a particular focus on the U.S., Canada, Europe, Japan, and South Korea. On average, background  $O_3$  concentrations in China (41.4  $\pm$  12.2 ppb) are slightly higher than those observed in the U.S. (35.7  $\pm$  14.0 ppb) (Chan and Vet, 2010; Dolwick et al., 2015; Emery et al., 2012; Fiore et al., 2003, 2002a; Hirsch et al., 1996; Parrish et al., 2009; Parrish and Ennis, 2019; Steiner et al., 2010; Vingarzan, 2004; Yan et al., 2021; Zhang et al., 2011) and Europe (34.2  $\pm$  10.3 ppb) (Auvray and Bey, 2005; Brönnimann et al., 2000; Kalabokas et al., 2000; Naja et al., 2003; Parrish et al., 2009; Scheel et al., 1997; Vecchi and Valli, 1998; Vingarzan, 2004; Wilson et al., 2012). This suggests that although developed regions have made significant progress in controlling anthropogenic  $O_3$  precursors, background  $O_3$  remains a major concern due to various regional factors such as higher emissions, industrial activity, and specific atmospheric conditions (Huang et al., 2015). In contrast, background  $O_3$  levels in China are significantly higher than those observed in Canada (26.9  $\pm$  7.4 ppb) (Chan and Vet, 2010; Vingarzan, 2004), which is likely due to Canada's lower industrial activity, less dense population, and colder climate that limits the photochemical processes necessary for  $O_3$  formation.

When comparing China to other East Asian regions, the background  $O_3$  concentration is slightly higher than in South Korea (38.8  $\pm$  11.74 ppb) (Ghim and Chang, 2000; Kim et al., 2023; Lam and Cheung, 2022; Lee and Park, 2022; Yeo and Kim, 2021), but marginally lower than in Japan (45.4  $\pm$  23.2 ppb) (Akimoto et al., 2015; Lam and Cheung, 2022; Sunwoo et al., 1994; Tsutsumi et al., 1994). Detailed

information on the data, including a breakdown of regional and temporal distributions, is provided Table S4. Notably, East Asian regions, including China, South Korea, and Japan, typically exhibit background O<sub>3</sub> levels that are 3–20 ppb higher than those observed in Europe, the U.S., and Canada. This regional disparity is attributable to a combination of factors, including the region's warm climate, high solar radiation, and the presence of industrialized areas that emit large quantities of O<sub>3</sub> precursors. These factors collectively enhance photochemical O<sub>3</sub> (Lee et al., 2021; Li et al., 2016; Nagashima et al., 2010; Yamaji et al., 2006). Furthermore, complex regional airflow patterns, including transboundary transport and local atmospheric dynamics, promote the accumulation of background O<sub>3</sub>, especially in densely populated urban centers. These findings underscore the critical need for regional cooperation in addressing O<sub>3</sub> pollution in East Asia, where transboundary influences and shared atmospheric conditions complicate the management of background O<sub>3</sub> levels.

A more granular regional comparison reveals notable differences in background O<sub>3</sub> concentrations among various regions of both China and the U.S. Specifically, the difference in background O<sub>3</sub> concentrations between central and western China (including NWC and SWC) reaches 10 ppb, while the discrepancy between the Eastern and Western U.S. is as high as 13 ppb. Western China and the Western U.S. exhibit higher background O<sub>3</sub> levels. In particular, the Los Angeles area in the Western U.S. reports background O<sub>3</sub> levels as high as 62 ppb (Parrish and Ennis, 2019), a phenomenon attributed to the region's combination of intense ultraviolet radiation, low humidity, and favorable atmospheric conditions for O<sub>3</sub> formation. Similarly, the higher altitudes of western China enhance its susceptibility to stratospheric transport, which contributes to elevated O<sub>3</sub> concentrations. The Western U.S. is similarly influenced by trans-Pacific atmospheric transport, further exacerbating O<sub>3</sub> levels.

In contrast to the significant regional differences observed in China and the U.S., background O<sub>3</sub> concentrations in Canada and Europe exhibit relatively small variations, typically ranging from 4 to 7 ppb. The limited variation in Canada can be attributed to factors such as its low population density, minimal industrial activity, and expansive natural vegetation, all of which, coupled with its cold climate, limit O<sub>3</sub> production. In Europe, the relatively smaller regional differences are likely as a result of effective transnational air quality management and stringent pollution control policies, which have successfully

minimized disparities in O<sub>3</sub> concentrations across the continent. The relatively uniform air quality management frameworks in these regions have helped mitigate large-scale emissions and reduce regional discrepancies in background O<sub>3</sub> levels (Miranda et al., 2015; Næss, 2004; Rodrigues et al., 2021; Xu et al., 2019).

Figure 8: Average background O<sub>3</sub> concentrations in the U.S., Canada, Europe, South Korea, Japan, and China.

## 6 Conclusions and perspectives

Background  $O_3$  concentrations are critical for understanding  $O_3$  pollution, as they represent the baseline level of  $O_3$  even in the absence of local anthropogenic emissions. These concentrations determine the maximum achievable reduction in  $O_3$  through the mitigation of anthropogenic precursor emissions, making accurate estimates crucial for effective air quality management and setting realistic pollution control targets. This study provides a comprehensive review of the definition and estimation methods for background  $O_3$  concentrations, with a focus on recent advances in regional research in China. Our findings reveal an average background  $O_3$  concentration of  $41.4 \pm 12.2$  ppb in China, which accounts for 79% of the tropospheric MDA8  $O_3$ . Notable spatial variations are observed, with the highest levels in Northwest China (NWC,  $48.2 \pm 8.3$  ppb) and the lowest in Northeast China (NEC,  $33.1 \pm 5.7$  ppb), alongside an upward national trend reflecting growing  $O_3$  pollution. Despite progress in estimation methods, discrepancies persist across the four estimation methods, with the in situ measurement

estimation method and statistical analysis estimation method yielding higher values, while the integrated methods offers lower but more consistent estimates. Compared to other regions, East Asia, including China, South Korea, and Japan, experiences background O<sub>3</sub> levels 3–20 ppb higher than the U.S., Canada, and Europe. This highlights the region-specific atmospheric conditions and pollution characteristics, and the imperative of addressing background O<sub>3</sub> pollution within a global framework.

Although substantial progress has been made in estimating background O<sub>3</sub> over recent decades, considerable challenges remain due to the complexity of its sources and the multitude of influencing factors, particularly in the context of global climate changes and transboundary pollution. Future research should prioritize several key areas to advance the understanding and management of background O<sub>3</sub>:

## 6.1 Accurate quantification of background ozone sources and processes

Natural emissions, long-range transport, and stratosphere–troposphere exchange (STE) are key drivers of background O<sub>3</sub> concentrations; however, significant uncertainties remain in quantifying their individual contributions. To improve our understanding and predictive capabilities, future research must prioritize the refinement of quantification methods for these sources and processes. For instance, the variability of natural emissions, particularly from BVOCs and lightning, remains inadequately characterized across different climatic conditions. In addition, STE represents another critical but poorly understood source of background O<sub>3</sub>, with studies indicating significant seasonal and regional variations in its contribution (Lu et al., 2019a; Xie et al., 2017). Despite the critical importance of these processes, existing models often encounter difficulties in accurately simulating natural emissions and STE, primarily due to limitations in model structures and parameterization (Auvray and Bey, 2005; Griffiths et al., 2021; Huang et al., 2024; Koo et al., 2010). As a result, the accuracy of model predictions for background O<sub>3</sub> concentrations is compromised, resulting in increased uncertainties that hinder effective policy planning and air quality management.

## 6.2 Development of integrated methods techniques

Single method approaches for estimating background  $O_3$  concentrations have inherent limitations, as they often fail to capture the full spectrum of factors influencing  $O_3$  levels. For example, while numerical

models provide valuable insights, they frequently underestimate actual O<sub>3</sub> concentrations due to simplifications in chemical processes and uncertainties in input data. In contrast, statistical analysis estimation methods are heavily dependent on the availability and representativeness of observational data, which can be sparse or biased, particularly in regions with limited monitoring networks. These limitations highlight the necessity for more integrated approaches that combine the strengths of different methods.

In this context, the development of integrated methods techniques presents a promising approach to improve background O<sub>3</sub> estimation. By integrating observational data, statistical analysis, and numerical results, integrated methods estimation can mitigate the inherent limitations of each individual method. For example, data assimilation techniques, which combine model outputs with real-time observational data, have been shown to improve both spatial and temporal resolution, yielding more accurate and robust O<sub>3</sub> estimates (Skipper et al., 2021; Sun et al., 2024). Additionally, the integration of high-resolution regional models with long-term observational datasets can significantly enhance spatiotemporal coverage of background O<sub>3</sub> estimates, enabling precise characterization of O<sub>3</sub> variability across diverse geographic scales, from urban centers to remote rural areas. Recent advancements in machine learning-based fusion methods further extend the potential of data integration by uncovering nonlinear relationships among multiple data sources, thereby improving estimation accuracy. These approaches can also account for complex interactions between meteorological conditions, emission sources, and atmospheric chemistry, which are often challenging to capture using traditional methods. Given the potential of integrated methods techniques to provide more accurate and comprehensive background O<sub>3</sub> estimates, future research should prioritize their continued development and validation. Such efforts will improve the precision and reliability of background O<sub>3</sub> estimates, thereby enhancing our understanding of regional O<sub>3</sub> pollution dynamics and supporting the development of more effective air quality management strategies.

#### 6.3 Fostering international collaboration on long-range pollution transport

As air quality standards for O<sub>3</sub> become increasingly stringent, background O<sub>3</sub> concentrations have emerged as a critical challenge for many countries in achieving regulatory targets. This issue is

particularly pronounced in regions impacted by both local and transboundary pollution, where efforts to reduce domestic emissions may not fully address the underlying drivers of elevated background O<sub>3</sub> levels. For instance, studies conducted in the U.S. have demonstrated that despite substantial reductions in local emissions of O<sub>3</sub> precursors, background O<sub>3</sub> concentrations in some areas remain persistently high (Cooper et al., 2012; Huang et al., 2015). This phenomenon is partly attributed to long-range transport of pollutants, including O<sub>3</sub> precursors, from distant regions, often spanning international borders and even continents (Cynthia Lin et al., 2000; Dentener et al., 2010). Such transboundary pollution underscores the need for comprehensive international cooperation to effectively mitigate the challenges posed by background O<sub>3</sub>.

International collaboration is therefore essential for tackling the elevated background O<sub>3</sub>. To this end, fostering transboundary emission reduction agreements between countries and regions can play a pivotal role in curbing the long-range transport of O<sub>3</sub> and its precursors. Moreover, strengthening the global background O<sub>3</sub> monitoring network, particularly in remote regions and marine stations, would significantly enhance the capacity for real-time monitoring of background O<sub>3</sub> levels on a global scale.

## 6.4 Strengthening research on the interaction between background ozone and climate change

The impact of climate change on background O<sub>3</sub> concentrations represents a critical area for future research, with profound implications for air quality management and public health. Climate change is expected to affect background O<sub>3</sub> levels through multiple interconnected mechanisms. For example, rising temperatures and altered precipitation patterns are expected to affect natural emissions, such as BVOCs emissions from forests and NO<sub>x</sub> emissions from soil, both of which are particularly sensitive to climatic factors like temperature and humidity. These changes would, in turn, influence regional background O<sub>3</sub> levels. Beyond these direct emission impacts, climate change is likely to modify atmospheric circulation patterns, thereby affecting the long-range transport of atmospheric pollutants and the spatial distribution of background O<sub>3</sub>. Alterations in wind patterns and monsoon systems, for example, could significantly alter the transport of O<sub>3</sub> and its precursors over large distances, thereby exacerbating regional background O<sub>3</sub> levels, especially in areas downwind of major pollution sources (Collins et al., 2003; Sonwani et al., 2016; Sudo et al., 2003; Wu et al., 2008). Consequently, future research should

| 910 | prioritize understanding the dynamic interplay between climate change and background O <sub>3</sub>       |
|-----|-----------------------------------------------------------------------------------------------------------|
| 911 | concentrations to improve predictive models and inform effective air quality management strategies.       |
| 912 |                                                                                                           |
| 913 | Author contributions                                                                                      |
| 914 | CC collected the data.CC conducted the data analysis and prepared the draft with the support of WC, LG    |
| 915 | and YW. XD, XW, and MS contributed to the revision of the paper.                                          |
| 916 |                                                                                                           |
| 917 | Competing interests                                                                                       |
| 918 | The authors declare that they have no conflict of interest.                                               |
| 919 |                                                                                                           |
| 920 | Acknowledgments                                                                                           |
| 921 | This study was supported by the National Key Research and Development Program of China                    |
| 922 | (2023YFC3706205), the National Natural Science Foundation of China (42375109, 42121004), the              |
| 923 | Guangzhou Basic and Applied Basic Research Foundation (2024A04J0781), Guangdong Provincial                |
| 924 | General Colleges and Universities Innovation Team Project (Natural Science) (2024KCXTD004),               |
| 925 | Special Support Plan for High-Level Talents of Guangdong Province (2023JC07L057), the Project             |
| 926 | Commissioned by Foshan Ecological Environment Monitoring Station of Guangdong Province                    |
| 927 | (GZGK24P047C0239Z) and the high-performance computing platform of Jinan University.                       |
|     |                                                                                                           |
| 928 | References                                                                                                |
| 929 | Akimoto, H., Mori, Y., Sasaki, K., Nakanishi, H., Ohizumi, T., and Itano, Y.: Analysis of monitoring data |
| 930 | of ground-level ozone in Japan for long-term trend during 1990–2010: Causes of temporal and spatial       |
| 931 | variation, Atmos. Environ., 102, 302–310, https://doi.org/10.1016/j.atmosenv.2014.12.001, 2015.           |
| 932 | Altshuller, A. P. and Lefohn, A. S.: Background ozone in the planetary boundary layer over the United     |
| 933 | States, J. Air Waste Manage., 46, 134–141, https://doi.org/10.1080/10473289.1996.10467445, 1996.          |
| 934 | Auvray, M. and Bey, I.: Long-range transport to Europe: Seasonal variations and implications for the      |
| 935 | Furonean azone hudget I Geophys Res Atmos 110 1 22 https://doi.org/10.1020/2004ID005503                   |

- 2005.
- Berlin, S. R., Langford, A. O., Estes, M., Dong, M., and Parrish, D. D.: Magnitude, decadal changes, and
- impact of regional background ozone transported into the Greater Houston, Texas, area, Environ. Sci.
- Technol., 47, 13985–13992, https://doi.org/10.1021/es4037644, 2013.
- Breiman, L.: Random forests, Mach. Learn., 45, 5–32, https://doi.org/10.1023/A:1010933404324, 2001.
- Brönnimann, S., Schuepbach, E., Zanis, P., Buchmann, B., and Wanner, H.: A climatology of regional
- background ozone at different elevations in Switzerland (1992-1998), Atmos. Environ., 34, 5191-
- 5198, https://doi.org/10.1016/S1352-2310(00)00193-X, 2000.
- Chan, C. Y., Chan, L. Y., and Harris, J. M.: Urban and background ozone trend in 1984-1999 at
- subtropical Hong Kong, South China, Ozone-Sci. Eng., 25, 513-522,
- https://doi.org/10.1080/01919510390481829, 2003.
- Chan, E. and Vet, R. J.: Baseline levels and trends of ground level ozone in Canada and the United States,
- Atmos. Chem. Phys., 10, 8629–8647, https://doi.org/10.5194/acp-10-8629-2010, 2010.
- Ohen, J.: Spatial and temporal variation of ozone concentration and its influencing factors from 2017 to
- 2020 in Northeast China, M.S. theses, Harbin Normal University, 40 pp., 2024.
- Chen, W., Guenther, A. B., Shao, M., Yuan, B., Jia, S., Mao, J., Yan, F., Krishnan, P., and Wang, X.:
- Assessment of background ozone concentrations in China and implications for using region-specific
- volatile organic compounds emission abatement to mitigate air pollution, Environ. Pollut., 305,
- 119254, https://doi.org/10.1016/j.envpol.2022.119254, 2022.
- Chen, X.: Analysis of total ozone distribution and its influencing factors in western China based on
- satellite remote sensing, M.S. theses, Northwest Normal University, 55 pp., 2020.
- Cheng, Y., Wang, Y., Zhang, Y., Crawford, J. H., Diskin, G. S., Weinheimer, A. J., and Fried, A.: Estimator
- of surface ozone using formaldehyde and carbon monoxide concentrations over the eastern United
- States in summer, J. Geophys. Res.-Atmos., 123, 7642–7655, https://doi.org/10.1029/2018JD028452,
- 2018.
- Collins, W. J., Derwent, R. G., Garnier, B., Johnson, C. E., Sanderson, M. G., and Stevenson, D. S.:
- Effect of stratosphere-troposphere exchange on the future tropospheric ozone trend, J. Geophys. Res.-

- Atmos., 108, 1–10, https://doi.org/10.1029/2002JD002617, 2003.
- Colombi, N. K., Jacob, D. J., Yang, L. H., Zhai, S., Shah, V., Grange, S. K., Yantosca, R. M., Kim, S.,
- and Liao, H.: Why is ozone in South Korea and the Seoul metropolitan area so high and increasing?,
- Atmos. Chem. Phys., 23, 4031–4044. https://doi.org/10.5194/acp-23-4031-2023, 2023.
- Cooper, O. R., Gao, R. S., Tarasick, D., Leblanc, T., and Sweeney, C.: Long-term ozone trends at rural
- ozone monitoring sites across the United States, 1990–2010, J. Geophys. Res.-Atmos., 117, 1–24,
- https://doi.org/10.1029/2012JD018261, 2012.
- Crutzen, P. J.: Photochemical reactions initiated by and influencing ozone in unpolluted tropospheric air,
- Tellus, 26, 47–57, https://doi.org/10.3402/tellusa.v26i1-2.9736, 1974.
- Cynthia Lin, C., Jacob, D. J., Munger, J. W., and Fiore, A. M.: Increasing background ozone in surface
- air over the United States, Geophys. Res. Lett., 27, 3465–3468,
- https://doi.org/10.1029/2000GL011762, 2000.
- Dentener, F., Keating, T., and Akimoto, H. (Eds.): Hemispheric transport of air pollution 2010: Part A:
- Ozone and particulate matter, United Nations, New York and Geneva, 278 pp., ISBN 9789211170436,
- 2010.
- Ding, A. and Wang, T.: Influence of stratosphere-to-troposphere exchange on the seasonal cycle of
- surface ozone at Mount Waliguan in western China, Geophys. Res. Lett., 33, 1-4,
- https://doi.org/10.1029/2005GL024760, 2006.
- Dolwick, P., Akhtar, F., Baker, K. R., Possiel, N., Simon, H., and Tonnesen, G.: Comparison of
- background ozone estimates over the western United States based on two separate model
- methodologies, Atmos. Environ., 109, 282–296, https://doi.org/10.1016/j.atmosenv.2015.01.005,
- 2015.
- Duc, H., Azzi, M., Wahid, H., and Ha, Q. P.: Background ozone level in the Sydney Basin: Assessment
- and trend analysis, Int. J. Climatol., 33, 2298–2308, https://doi.org/10.1002/joc.3595, 2013.
- Emery, C., Jung, J., Downey, N., Johnson, J., Jimenez, M., Yarwood, G., and Morris, R.: Regional and
- global modeling estimates of Policy Relevant Background ozone over the United States, Atmos.
- Environ., 47, 206–217, https://doi.org/10.1016/j.atmosenv.2011.11.012, 2012.

- European Parliament and Council: Directive on ambient air quality and cleaner air for Europe,
- 2008/50/EC, https://eur-lex.europa.eu/eli/dir/2008/50/oj (last access: 1 August 2025), 2008.
- Fiore, A., Jacob, D. J., Liu, H., Yantosca, R. M., Fairlie, T. D., and Li, Q.: Variability in surface ozone
- background over the United States: Implications for air quality policy, J. Geophys. Res.-Atmos., 108,
- 1–19, https://doi.org/10.1029/2003JD003855, 2003.
- Fiore, A. M., Jacob, D. J., Bey, I., Yantosca, R. M., Field, B. D., Fusco, A. C., and Wilkinson, J. G.:
- Background ozone over the United States in summer: Origin, trend, and contribution to pollution
- episodes, J. Geophys. Res.-Atmos., 107, 1–25, https://doi.org/10.1029/2001JD000982, 2002a.
- Fiore, A. M., Jacob, D. J., Field, B. D., Streets, D. G., Fernandes, S. D., and Jang, C.: Linking ozone
- pollution and climate change: The case for controlling methane, Geophys. Res. Lett., 29, 25-1–25-4,
- https://doi.org/10.1029/2002GL015601, 2002b.
- Fiore, A. M., Oberman, J. T., Lin, M. Y., Zhang, L., Clifton, O. E., Jacob, D. J., Naik, V., Horowitz, L.
- W., Pinto, J. P., and Milly, G. P.: Estimating North American background ozone in U.S. surface air
- with two independent global models: Variability, uncertainties, and recommendations, Atmos.
- Environ., 96, 284–300, https://doi.org/10.1016/j.atmosenv.2014.07.045, 2014.
- Galbally, I. E., Miller, A. J., Hoy, R. D., Ahmet, S., Joynt, R. C., and Attwood, D.: Surface ozone at rural
- sites in the Latrobe Valley and Cape Grim, Australia, Atmos. Environ. (1967), 20, 2403–2422,
- https://doi.org/10.1016/0004-6981(86)90071-5, 1986.
- Gao, J., Wang, T., Ding, A., and Liu, C.: Observational study of ozone and carbon monoxide at the
- summit of Mount Tai (1534m a.s.l.) in central-eastern China, Atmos. Environ., 39, 4779–4791,
- https://doi.org/10.1016/j.atmosenv.2005.04.030, 2005.
- Geng, G., Liu, Y., Liu, Y., Liu, S., Cheng, J., Yan, L., Wu, N., Hu, H., Tong, D., Zheng, B., Yin, Z., He,
- 1012 K., and Zhang, Q.: Efficacy of China's clean air actions to tackle PM<sub>2.5</sub> pollution between 2013 and
- 2020, Nat. Geosci., 17, 987–994, https://doi.org/10.1038/s41561-024-01540-z, 2024.
- Ghim, Y. S. and Chang, Y.: Characteristics of ground-level ozone distributions in Korea for the period of
- 1990–1995, J. Geophys. Res.-Atmos., 105, 8877–8890, https://doi.org/10.1029/1999JD901179, 2000.
- Griffiths, P. T., Murray, L. T., Zeng, G., Shin, Y. M., Abraham, N. L., Archibald, A. T., Deushi, M.,

- Emmons, L. K., Galbally, I. E., Hassler, B., Horowitz, L. W., Keeble, J., Liu, J., Moeini, O., Naik, V.,
- O'Connor, F. M., Oshima, N., Tarasick, D., Tilmes, S., Turnock, S. T., Wild, O., Young, P. J., and Zanis,
- P.: Tropospheric ozone in CMIP6 simulations, Atmos. Chem. Phys., 21, 4187–4218,
- https://doi.org/10.5194/acp-21-4187-2021, 2021.
- Guo, J. J., Fiore, A. M., Murray, L. T., Jaffe, D. A., Schnell, J. L., Moore, C. T., and Milly, G. P.: Average
- versus high surface ozone levels over the continental USA: Model bias, background influences, and
- interannual variability, Atmos. Chem. Phys., 18, 12123–12140, https://doi.org/10.5194/acp-18-12123-
- 2018, 2018.
- Haagen-Smit, A. J.: Chemistry and physiology of Los Angeles Smog, Ind. Eng. Chem., 44, 1342–1346,
- https://doi.org/10.1021/ie50510a045, 1952.
- Han, H., Liu, J., Yuan, H., Wang, T., Zhuang, B., and Zhang, X.: Foreign influences on tropospheric
- ozone over East Asia through global atmospheric transport, Atmos. Chem. Phys., 19, 12495–12514.
- https://doi.org/10.5194/acp-19-12495-2019, 2019.
- He, C., Mu, H., Yang, L., Wang, D., Di, Y., Ye, Z., Yi, J., Ke, B., Tian, Y., and Hong, S.: Spatial variation
- of surface ozone concentration during the warm season and its meteorological driving factors in China,
- Environm. Sci., 42, 4168–4179, https://doi.org/10.13227/j.hjkx.202009228, 2021.
- He, C., Wu, Q., Li, B., Liu, J., Gong, X., and Zhang, L.: Surface ozone pollution in China: Trends,
- exposure risks, and drivers, Front. Public Health, 11, 1131753,
- https://doi.org/10.3389/fpubh.2023.1131753, 2023.
- Hirsch, A. I., Munger, J. W., Jacob, D. J., Horowitz, L.W., and Goldstein, A. H.: Seasonal variation of
- the ozone production efficiency per unit NO<sub>x</sub> at Harvard Forest, Massachusetts, J. Geophys. Res.-
- Atmos., 101, 12659–12666, 1996.
- Hogrefe, C., Liu, P., Pouliot, G., Mathur, R., Roselle, S., Flemming, J., Lin, M., and Park, R. J.: Impacts
- of different characterizations of large-scale background on simulated regional-scale ozone over the
- continental United States, Atmos. Chem. Phys., 18, 3839–3864, https://doi.org/10.5194/acp-18-3839-
- 2018, 2018.
- Hosseinpour, F., Kumar, N., Tran, T., and Knipping, E.: Using machine learning to improve the estimate

- of U.S. background ozone, Atmos. Environ., 316, 120145,
- https://doi.org/10.1016/j.atmosenv.2023.120145, 2024.
- Hu, C., Kang, P., Wu, K., Zhang, X., Wang, S., Wang, Z., Ouyang, Z., Zeng, S., and Wei, X.: Study of
- the spatial and temporal distribution of ozone and its influence factors over Sichuan Basin based on
- generalized additive model, Acta Scien. Circum., 39, 809–820,
- https://doi.org/10.13671/j.hjkxxb.2018.0444, 2019.
- Huang, L., Zhao, X., Chen, C., Tan, J., Li, Y., Chen, H., Wang, Y., Li, L., Guenther, A., and Huang, H.:
- Uncertainties of biogenic VOC emissions caused by land cover data and implications on ozone
- mitigation strategies for the Yangtze River Delta region, Atmos. Environ., 337, 120765,
- https://doi.org/10.1016/j.atmosenv.2024.120765, 2024.
- Huang, M., Bowman, K. W., Carmichael, G. R., Lee, M., Chai, T., Spak, S. N., Henze, D. K., Darmenov,
- 1055 A. S., and da Silva, A. M.: Improved western U.S. background ozone estimates via constraining
- nonlocal and local source contributions using Aura TES and OMI observations, J. Geophys. Res.-
- Atmos., 120, 3572–3592, https://doi.org/10.1002/2014JD022993, 2015.
- Itano, Y., Bandow, H., Takenaka, N., Saitoh, Y., Asayama, A., and Fukuyama, J.: Impact of NO<sub>x</sub> reduction
- on long-term ozone trends in an urban atmosphere, Sci. Total Environ., 379, 46-55,
- https://doi.org/10.1016/j.scitotenv.2007.01.079, 2007.
- Jackson, R. B., Saunois, M., Martinez, A., Canadell, J. G., Yu, X., Li, M., Poulter, B., Raymond, P. A.,
- Regnier, P., Ciais, P., Davis, S. J., and Patra, P. K.: Human activities now fuel two-thirds of global
- methane emissions, Environ. Res. Lett., 19, 101002, https://doi.org/10.1088/1748-9326/ad6463, 2024.
- Jacob, D. J., Logan, J. A., and Murti, P. P.: Effect of rising Asian emissions on surface ozone in the United
- States, Geophys. Res. Lett., 26, 2175–2178, https://doi.org/10.1029/1999GL900450, 1999.
- Jaffe, D. A., Cooper, O. R., Fiore, A. M., Henderson, B. H., Tonnesen, G. S., Russell, A. G., Henze, D.
- 1067 K., Langford, A. O., Lin, M., and Moore, T.: Scientific assessment of background ozone over the U.S.:
- Implications for air quality management, Elementa-Sci. Anthrop., 6, 56,
- https://doi.org/10.1525/elementa.309, 2018.
- Jenkin, M. E.: Trends in ozone concentration distributions in the UK since 1990: Local, regional and

- global influences, Atmos. Environ., 42, 5434–5445, https://doi.org/10.1016/j.atmosenv.2008.02.036,
- 2008.
- Jolliffe, I.: Principal Component Analysis, in: Encyclopedia of statistics in behavioral science, edited by:
- Everitt, B. S. and Howell, D. C., John Wiley and Sons, Ltd., Chichester, 1580–1584,
- https://doi.org/10.1002/0470013192.bsa501, 2005.
- Kalabokas, P. D., Viras, L. G., Bartzis, J. G., and Repapis, C. C.: Mediterranean rural ozone
- characteristics around the urban area of Athens, Atmos. Environ., 34, 5199–5208,
- https://doi.org/10.1016/S1352-2310(00)00298-3, 2000.
- Kashinath, K., Mustafa, M., Albert, A., Wu, J., Jiang, C., Esmaeilzadeh, S., Azizzadenesheli, K., Wang,
- R., Chattopadhyay, A., Singh, A., Manepalli, A., Chirila, D., Yu, R., Walters, R., White, B., Xiao, H.,
- Tchelepi, H. A., Marcus, P., Anandkumar, A., Hassanzadeh, P., and Prabhat: Physics-informed machine
- learning: Case studies for weather and climate modelling, Philos. T. R. Soc. A., 379, 20200093,
- https://doi.org/10.1098/rsta.2020.0093, 2021.
- Kemball-Cook, S., Parrish, D., Ryerson, T., Nopmongcol, U., Johnson, J., Tai, E., and Yarwood, G.:
- Contributions of regional transport and local sources to ozone exceedances in Houston and Dallas:
- Comparison of results from a photochemical grid model to aircraft and surface measurements, J.
- Geophys. Res.-Atmos., 114, 1–14, https://doi.org/10.1029/2008JD010248, 2009.
- Kim, S., Kim, K., Jeong, Y., Seo, S., Park, Y., and Kim, J.: Changes in surface ozone in South Korea on
- diurnal to decadal timescales for the period of 2001–2021, Atmos. Chem. Phys., 23, 12867–12886,
- https://doi.org/10.5194/acp-23-12867-2023, 2023.
- Kirschke, S., Bousquet, P., Ciais, P., Saunois, M., Canadell, J. G., Dlugokencky, E. J., Bergamaschi, P.,
- Bergmann, D., Blake, D. R., Bruhwiler, L., Cameron-Smith, P., Castaldi, S., Chevallier, F., Feng, L.,
- Fraser, A., Heimann, M., Hodson, E. L., Houweling, S., Josse, B., Fraser, P. J., Krummel, P. B.,
- Lamarque, J. F., Langenfelds, R. L., Le Quere, C., Naik, V., O'Doherty, S., Palmer, P. I., Pison, I.,
- Plummer, D., Poulter, B., Prinn, R. G., Rigby, M., Ringeval, B., Santini, M., Schmidt, M., Shindell, D.
- T., Simpson, I. J., Spahni, R., Steele, L. P., Strode, S. A., Sudo, K., Szopa, S., van der Werf, G. R.,
- Voulgarakis, A., van Weele, M., Weiss, R. F., Williams, J. E., and Zeng, G.: Three decades of global

- methane sources and sinks, Nat. Geosci., 6, 813–823, https://doi.org/10.1038/ngeo1955, 2013.
- Koo, B., Chien, C., Tonnesen, G., Morris, R., Johnson, J., Sakulyanontvittaya, T., Piyachaturawat, P., and
- Yarwood, G.: Natural emissions for regional modeling of background ozone and particulate matter
- and impacts on emissions control strategies, Atmos. Environ., 44, 2372–2382,
- https://doi.org/10.1016/j.atmosenv.2010.02.041, 2010.
- Lam, Y. F. and Cheung, H. M.: Investigation of Policy Relevant Background (PRB) ozone in East Asia,
- Atmosphere-Basel, 13, 723, https://doi.org/10.3390/atmos13050723, 2022.
- Langford, A. O., Senff, C. J., Banta, R. M., Hardesty, R. M., Alvarez II, R. J., Sandberg, S. P., and Darby,
- 1106 L. S.: Regional and local background ozone in Houston during Texas Air Quality Study 2006, J.
- Geophys. Res.-Atmos., 114, 1–12, https://doi.org/10.1029/2008JD011687, 2009.
- Lassey, K. R., Lowe, D. C., and Manning, M. R.: The trend in atmospheric methane \delta 13C and
- implications for isotopic constraints on the global methane budget, Global Biogeochem. Cy., 14, 41–
- 49, https://doi.org/10.1029/1999GB900094, 2000.
- Lee, H., Chang, L., Jaffe, D. A., Bak, J., Liu, X., Abad, G. G., Jo, H., Jo, Y., Lee, J., and Kim, C.: Ozone
- continues to increase in East Asia despite decreasing NO<sub>2</sub>: Causes and abatements, Remote Sens.-
- Basel, 13, 2177, https://doi.org/10.3390/rs13112177, 2021.
- Lee, H. and Park, R. J.: Factors determining the seasonal variation of ozone air quality in South Korea:
- Regional background versus domestic emission contributions, Environ. Pollut., 308, 119645,
- https://doi.org/10.1016/j.envpol.2022.119645, 2022.
- Lee, Y. C., Shindell, D. T., Faluvegi, G., Wenig, M., Lam, Y. F., Ning, Z., Hao, S., and Lai, C. S.: Increase
- of ozone concentrations, its temperature sensitivity and the precursor factor in South China. Tellus B:
- Chem. Phy. Meteorol., 66, 23455, https://doi.org/10.3402/tellusb.v66.23455, 2014.
- Lefohn, A. S., Emery, C., Shadwick, D., Wernli, H., Jung, J., and Oltmans, S. J.: Estimates of background
- surface ozone concentrations in the United States based on model-derived source apportionment,
- Atmos. Environ., 84, 275–288, https://doi.org/10.1016/j.atmosenv.2013.11.033, 2014.
- Lelieveld, J., Crutzen, P. J., and Dentener, F. J.: Changing concentration, lifetime and climate forcing of
- atmospheric methane, Tellus B, 50, 128–150, https://doi.org/10.1034/j.1600-0889.1998.t01-1-00002.x,

- 1998.
- Li, J., Yang, W., Wang, Z., Chen, H., Hu, B., Li, J., Sun, Y., Fu, P., and Zhang, Y.: Modeling study of
- surface ozone source-receptor relationships in East Asia, Atmos. Res., 167, 77-88,
- https://doi.org/10.1016/j.atmosres.2015.07.010, 2016.
- Li, K., Jacob, D. J., Liao, H., Shen, L., Zhang, Q., and Bates, K. H.: Anthropogenic drivers of 2013–2017
- trends in summer surface ozone in China, P. Natl. A. Sci. U.S.A., 116, 422-427,
- https://doi.org/10.1073/pnas.1812168116, 2019.
- Li, N., He, Q., Greenberg, J., Guenther, A., Li, J., Cao, J., Wang, J., Liao, H., Wang, Q., and Zhang, Q.:
- Impacts of biogenic and anthropogenic emissions on summertime ozone formation in the Guanzhong
- Basin, China, Atmos. Chem. Phys., 18, 7489–7507, https://doi.org/10.5194/acp-18-7489-2018, 2018.
- Li, X., Liu, J., Mauzerall, D. L., Emmons, L. K., Walters, S., Horowitz, L. W., and Tao, S.: Effects of
- trans-Eurasian transport of air pollutants on surface ozone concentrations over western China, J.
- Geophys. Res.-Atmos., 119, 12338–12354, https://doi.org/10.1002/2014JD021936, 2014.
- Li, Y., Lau, A. K., Fung, J. C., Zheng, J. Y., Zhong, L. J., and Louie, P. K. K.: Ozone source apportionment
- (OSAT) to differentiate local regional and super-regional source contributions in the Pearl River Delta
- region, China, J. Geophys. Res.-Atmos., 117, 1–18, https://doi.org/10.1029/2011JD017340, 2012.
- Liu, H., Zhang, M., and Han, X.: A review of surface ozone source apportionment in China, Atmos.
- Oceanic Sci. Lett., 13, 470–484, https://doi.org/10.1080/16742834.2020.1768025, 2020.
- Liu, N., Lin, W., Ma, J., Xu, W., and Xu, X.: Seasonal variation in surface ozone and its regional
- characteristics at global atmosphere watch stations in China, J. Environ. Sci., 77, 291-302,
- https://doi.org/10.1016/j.jes.2018.08.009, 2019.
- Liu, N., Li, X., Ren, W., and Wan, R.: Influence of East Asian summer monsoon on ozone transport in
- eastern China, Trans. Atmos. Sci. (in Chinese), 44, 261–269,
- https://doi.org/10.13878/j.cnki.dqkxxb.20200706001, 2021.
- Liu, S. C., Trainer, M., Fehsenfeld, F. C., Parrish, D. D., Williams, E. J., Fahey, D. W., Hübler, G., and
- Murphy, P. C.: Ozone production in the rural troposphere and the implications for regional and global
- ozone distributions, J. Geophys. Res.-Atmos., 92, 4191–4207,

- https://doi.org/10.1029/JD092iD04p04191, 1987.
- Lu, X., Zhang, L., Chen, Y., Zhou, M., Zheng, B., Li, K., Liu, Y., Lin, J., Fu, T., and Zhang, Q.: Exploring
- 2016–2017 surface ozone pollution over China: Source contributions and meteorological influences,
- Atmos. Chem. Phys., 19, 8339–8361, https://doi.org/10.5194/acp-19-8339-2019, 2019a.
- Lu, X., Zhang, L., and Shen, L.: Meteorology and climate influences on tropospheric ozone: a review of
- natural sources, chemistry, and transport patterns, Curr. Pollut. Rep., 5, 238–260.
- https://doi.org/10.1007/s40726-019-00118-3, 2019b.
- Luo, Y., Peng, Q., Jin, H., Zhang, Q., Yin, W., Zeng, Y., Li, W., and Xiao, T.: The characteristics of ozone
- concentration of Hengyang and Heng Mountain background station, Environmental Monitoring in
- China, 35, 100–108, https://doi.org/10.19316/j.issn.1002-6002.2019.03.14, 2019.
- Luo, Y., Zhao, T., Meng, K., Hu, J., Yang, Q., Bai, Y., Yang, K., Fu, W., Tan, C., Zhang, Y., Zhang, Y.,
- and Li, Z.: A mechanism of stratospheric O<sub>3</sub> intrusion into the atmospheric environment: a case study
- of the North China Plain, Atmos. Chem. Phys., 24, 7013–7026, https://doi.org/10.5194/acp-24-7013-
- 2024, 2024.
- 1166 Ma, J., Zheng, X., and Xu, X.: Comment on "Why does surface ozone peak in summertime at Waliguan?"
- by Bin Zhu et al., Geophys. Res. Lett., 32, 1–2, https://doi.org/10.1029/2004GL021683, 2005.
- 1168 Ma, J., Yan, Y., Kong, S., Bai, Y., Zhou, Y., Gu, X., Song, A., and Tong Z.: Effectiveness of inter-regional
- collaborative emission reduction for ozone mitigation under local-dominated and transport-affected
- synoptic patterns, Environ. Sci. Pollut. Res., 31, 51774–51789, https://doi.org/10.1007/s11356-024-
- 34656-1, 2024.
- 1172 Ma, Z., Xu, J., Quan, W., Zhang, Z., Lin, W., and Xu, X.: Significant increase of surface ozone at a rural
- site, north of eastern China, Atmos. Chem. Phys., 16, 3969–3977, https://doi.org/10.5194/acp-16-
- 3969-2016, 2016.
- 1175 Ma, X., Huang, J., Hegglin, M. I., Jöckel, P., and Zhao, T.: Causes of growing middle-to-upper
- tropospheric ozone over the northwest Pacific region, Atmos. Chem. Phys., 25, 943-958,
- https://doi.org/10.5194/acp-25-943-2025, 2025.
- Mahmud, A., Tyree, M., Cayan, D., Motallebi, N., and Kleeman, M. J.: Statistical downscaling of climate

- change impacts on ozone concentrations in California, J. Geophys. Res.-Atmos., 113, 1-12,
- https://doi.org/10.1029/2007JD009534, 2008.
- McDonald-Buller, E. C., Allen, D. T., Brown, N., Jacob, D. J., Jaffe, D., Kolb, C. E., Lefohn, A. S.,
- Oltmans, S., Parrish, D. D., Yarwood, G., and Zhang, L.: Establishing Policy Relevant Background
- (PRB) ozone concentrations in the United States, Environ. Sci. Technol., 45, 9484–9497,
- https://doi.org/10.1021/es2022818, 2011.
- Miranda, A., Silveira, C., Ferreira, J., Monteiro, A., Lopes, D., Relvas, H., Borrego, C., and Roebeling,
- P.: Current air quality plans in Europe designed to support air quality management policies, Atmos.
- Pollut. Res., 6, 434–443, https://doi.org/10.5094/APR.2015.048, 2015.
- Næss, T.: The Effectiveness of the EU's ozone policy, Int. Environ. Agreem.-P, 4, 47-63,
- https://doi.org/10.1023/B:INEA.0000019051.06627.2e, 2004.
- Nagashima, T., Ohara, T., Sudo, K., and Akimoto, H.: The relative importance of various source regions
- on East Asian surface ozone, Atmos. Chem. Phys., 10, 11305–11322, https://doi.org/10.5194/acp-10-
- 11305-2010, 2010.
- Naja, M., Akimoto, H., and Staehelin, J.: Ozone in background and photochemically aged air over central
- Europe: Analysis of long-term ozonesonde data from Hohenpeissenberg and Payerne, J. Geophys.
- Res.-Atmos., 108, 1–11, https://doi.org/10.1029/2002JD002477, 2003.
- Ni, R., Lin, J., Yan, Y., and Lin, W.: Foreign and domestic contributions to springtime ozone over China,
- Atmos. Chem. Phys., 18, 11447–11469, https://doi.org/10.5194/acp-18-11447-2018, 2018.
- Nie, H., Niu, S., Wang, Z., Tang, J., and Zhao, Y.: Characteristic analysis of surface ozone over clean
- area in Qinghai-Xizang Plateau, Journal of Arid Meteorology, 22, 1–7, 2004.
- Nielsen-Gammon, J. W., Tobin, J., McNeel, A., and Li, G.: A conceptual model for eight-hour ozone
- exceedances in Houston, Texas Part I: Background ozone levels in eastern Texas, Center for
- Atmospheric Chemistry and the Environment, Texas A&M University, Open File Rep., 52 pp., 2005.
- Nopmongcol, U., Jung, J., Kumar, N., and Yarwood, G.: Changes in US background ozone due to global
- anthropogenic emissions from 1970 to 2020, Atmos. Environ., 140, 446–455,
- https://doi.org/10.1016/j.atmosenv.2016.06.026, 2016.

- Ou-Yang, C., Hsieh, H., Wang, S., Lin, N., Lee, C., Sheu, G., and Wang, J.: Influence of Asian continental
- outflow on the regional background ozone level in northern South China Sea, Atmos. Environ., 78,
- 144–153, https://doi.org/10.1016/j.atmosenv.2012.07.040, 2013.
- Ozone Pollution Control Committee of Chinese Society of Environmental Sciences: China blue book on
- prevention and control of atmospheric ozone pollution (2020), Science Press, Chinese mainland, 121
- pp., ISBN 9787030716644, 2022.
- Ozone Pollution Control Committee of Chinese Society of Environmental Sciences: China blue book on
- prevention and control of atmospheric ozone pollution (2023), Science Press, Chinese mainland, 164
- pp., ISBN 9787030781840, 2024.
- Parrish, D. D. and Ennis, C. A.: Estimating background contributions and US anthropogenic
- enhancements to maximum ozone concentrations in the northern US, Atmos. Chem. Phys., 19, 12587—
- 12605, https://doi.org/10.5194/acp-19-12587-2019, 2019.
- Parrish, D. D., Millet, D. B., and Goldstein, A. H.: Increasing ozone in marine boundary layer inflow at
- the west coasts of North America and Europe, Atmos. Chem. Phys., 9, 1303-1323,
- https://doi.org/10.5194/acp-9-1303-2009, 2009.
- Pfister, G. G., Walters, S., Emmons, L. K., Edwards, D. P., and Avise, J.: Quantifying the contribution of
- inflow on surface ozone over California during summer 2008, J. Geophys. Res.-Atmos., 118, 12282–
- 12299, https://doi.org/10.1002/2013JD020336, 2013.
- Prather, M. J., Holmes, C. D., and Hsu, J.: Reactive greenhouse gas scenarios: Systematic exploration of
- uncertainties and the role of atmospheric chemistry, Geophys. Res. Lett., 39, L09803,
- https://doi.org/10.1029/2012gl051440, 2012.
- Qin, L., Han, X., Zhang, M., and Liu, J.: Numerical simulation analysis on ozone source apportionment
- in the Yinchuan metropolitan area in summer, Climatic Environ. Res. (in Chinese), 28, 183–194,
- https://doi.org/10.3878/j.issn.1006-9585.2022.21186, 2023.
- Reid, N., Yap, D., and Bloxam, R.: The potential role of background ozone on current and emerging air
- issues: An overview, Air Qual. Atmos. Hlth., 1, 19–29, https://doi.org/10.1007/s11869-008-0005-z,
- 2008.

- Riley, M. L., Jiang, N., Duc, H. N., and Azzi, M.: Long-term trends in inferred continental background
- ozone in eastern Australia, Atmosphere-Basel, 14, 1104, https://doi.org/10.3390/atmos14071104, 2023.
- Rizos, K., Meleti, C., Kouvarakis, G., Mihalopoulos, N., and Melas, D.: Determination of the background
- pollution in the eastern Mediterranean applying a statistical clustering technique, Atmos. Environ.,
- 276, 119067, https://doi.org/10.1016/j.atmosenv.2022.119067, 2022.
- Rodrigues, V., Gama, C., Ascenso, A., Oliveira, K., Coelho, S., Monteiro, A., Hayes, E., and Lopes, M.:
- Assessing air pollution in European cities to support a citizen centered approach to air quality
- management, Sci. Total Environ., 799, 149311, https://doi.org/10.1016/j.scitotenv.2021.149311, 2021.
- Sahu, S. K., Liu, S., Liu, S., Ding, D., and Xing, J.: Ozone pollution in China: Background and
- transboundary contributions to ozone concentration & related health effects across the country, Sci.
- Total Environ., 761, 144131, https://doi.org/10.1016/j.scitotenv.2020.144131, 2021.
- Saunois, M., Bousquet, P., Poulter, B., Peregon, A., Ciais, P., Canadell, J. G., Dlugokencky, E. J., Etiope,
- G., Bastviken, D., Houweling, S., Janssens-Maenhout, G., Tubiello, F. N., Castaldi, S., Jackson, R. B.,
- Alexe, M., Arora, V. K., Beerling, D. J., Bergamaschi, P., Blake, D. R., Brailsford, G., Brovkin, V.,
- Bruhwiler, L., Crevoisier, C., Crill, P., Covey, K., Curry, C., Frankenberg, C., Gedney, N., Höglund-
- Isaksson, L., Ishizawa, M., Ito, A., Joos, F., Kim, H.-S., Kleinen, T., Krummel, P., Lamarque, J.-F.,
- Langenfelds, R., Locatelli, R., Machida, T., Maksyutov, S., McDonald, K. C., Marshall, J., Melton, J.
- R., Morino, I., Naik, V., O'Doherty, S., Parmentier, F.-J. W., Patra, P. K., Peng, C., Peng, S., Peters, G.
- P., Pison, I., Prigent, C., Prinn, R., Ramonet, M., Riley, W. J., Saito, M., Santini, M., Schroeder, R.,
- Simpson, I. J., Spahni, R., Steele, P., Takizawa, A., Thornton, B. F., Tian, H., Tohjima, Y., Viovy, N.,
- Voulgarakis, A., van Weele, M., van der Werf, G. R., Weiss, R., Wiedinmyer, C., Wilton, D. J.,
- Wiltshire, A., Worthy, D., Wunch, D., Xu, X., Yoshida, Y., Zhang, B., Zhang, Z., and Zhu, Q.: The
- global methane budget 2000–2012, Earth Syst. Sci. Data, 8, 697–751, https://doi.org/10.5194/essd-8-
- 697-2016, 2016.
- Scheel, H. E., Areskoug, H., Geiss, H., Gomiscek, B., Granby, K., Haszpra, L., Klasinc, L., Kley, D.,
- Laurila, T., Lindskog, A., Roemer, M., Schmitt, R., Simmonds, P., Solberg, S., and Toupance, G.: On
- the spatial distribution and seasonal variation of lower-troposphere ozone over Europe, J. Atmos.

- Chem., 28, 11–28, https://doi.org/10.1023/A:1005882922435, 1997.
- Shen, J., He, L., Cheng, P., Xie, M., Jiang, M., Chen, D., and Zhou, G.: Characteristics of ozone
- concentration variation in the northern background site of the Pearl River Delta, Ecology and
- Environmental Sciences, 28, 2006–2011, https://doi.org/10.16258/j.cnki.1674–5906.2019.10.010,
- 2019.
- Shin, H. J., Cho, K. M., Han, J. S., Kim, J. S., and Kim, Y. P.: The effects of precursor emission and
- background concentration changes on the surface ozone concentration over Korea, Aerosol Air Qual.
- Res., 12, 93–103, https://doi.org/10.4209/aagr.2011.09.0141, 2012.
- Sillman, S. and Samson, P. J.: Impact of temperature on oxidant photochemistry in urban, polluted rural
- and remote environments, J. Geophys. Res.-Atmos., 100, 11497–11508,
- https://doi.org/10.1029/94JD02146, 1995.
- Skipper, T. N., Hu, Y., Odman, M. T., Henderson, B. H., Hogrefe, C., Mathur, R., and Russell, A. G.:
- Estimating US background ozone using data fusion, Environ. Sci. Technol., 55, 4504–4512,
- https://doi.org/10.1021/acs.est.0c08625, 2021.
- Sonwani, S., Saxena, P., and Kulshrestha, U.: Role of global warming and plant signaling in BVOC
- emissions, in: Plant responses to air pollution, edited by: Kulshrestha, U. and Saxena, P., Springer,
- Singapore, 45–57, https://doi.org/10.1007/978-981-10-1201-3 5, 2016.
- Steiner, A. L., Davis, A. J., Sillman, S., Owen, R. C., Michalak, A. M., and Fiore, A. M.: Observed
- suppression of ozone formation at extremely high temperatures due to chemical and biophysical
- feedbacks, P. Natl. A. Sci. U.S.A., 107, 19685–19690, https://doi.org/10.1073/pnas.1008336107, 2010.
- Su, B.: Characteristics and impact factors of O<sub>3</sub> concentrations in mountain background region of East
- China, Environm. Sci., 34, 2519–2525, https://doi.org/10.13227/j.hjkx.2013.07.023, 2013.
- Sudo, K., Takahashi, M., and Akimoto, H.: Future changes in stratosphere-troposphere exchange and
- their impacts on future tropospheric ozone simulations, Geophys. Res. Lett., 30, 1-4,
- https://doi.org/10.1029/2003GL018526, 2003.
- Sun, L., Xue, L., Wang, T., Gao, J., Ding, A., Cooper, O. R., Lin, M., Xu, P., Wang, Z., Wang, X., Wen,
- L., Zhu, Y., Chen, T., Yang, L., Wang, Y., Chen, J., and Wang, W.: Significant increase of summertime

- ozone at Mount Tai in central eastern China, Atmos. Chem. Phys., 16, 10637–10650,
- https://doi.org/10.5194/acp-16-10637-2016, 2016.
- Sun, Z., Tan, J., Wang, F., Li, R., Zhang, X., Liao, J., Wang, Y., Huang, L., Zhang, K., Fu, J. S., and Li,
- 1290 L.: Regional background ozone estimation for China through data fusion of observation and simulation,
- Sci. Total Environ., 912, 169411, https://doi.org/10.1016/j.scitotenv.2023.169411, 2024.
- Sunwoo, Y., Carmichael, G. R., and Ueda, H.: Characteristics of background surface ozone in Japan,
- Atmos. Environ., 28, 25–37, https://doi.org/10.1016/1352-2310(94)90020-5, 1994.
- Thompson, T. M.: Background ozone: Challenges in science and policy, Congressional Research Service,
- Library of Congress, CRS Rep. R45482, 13 pp., 2019.
- Tsutsumi, Y., Zaizen, Y., and Makino, Y.: Tropospheric ozone measurement at the top of Mt. Fuji,
- Geophys. Res. Lett., 21, 1727–1730, https://doi.org/10.1029/94GL01107, 1994.
- U.S. EPA.: Air quality criteria for ozone and related photochemical oxidants (final report, 2006), U.S.
- Environmental Protection Agency, Washington, DC, EPA/600/R-05/004aF-cF, 2006.
- U.S. EPA.: Electronic Code of Federal Regulations, Title 40: Protection of Environment Chapter I
- (Environmental Protection Agency), Part 53: Ambient air monitoring reference and equivalent
- methods, https://www.ecfr.gov/current/title-40/chapter-I/part-53, (last access: 1 August 2025), 2011.
- U.S. EPA.: Review of the national ambient air quality standards for ozone: Policy assessment of scientific
- and technical information, U.S. Environmental Protection Agency, Research Triangle Park, North
- Carolina, EPA-452/R-07-007, 2007.
- Valdes, P. J., Beerling, D. J., and Johnson, C. E.: The ice age methane budget, Geophys. Res. Lett., 32,
- L02704, https://doi.org/10.1029/2004GL021004, 2005.
- Vecchi, R. and Valli, G.: Ozone assessment in the southern part of the Alps, Atmos. Environ., 33, 97-
- 109, https://doi.org/10.1016/S1352-2310(98)00133-2, 1998.
- Vingarzan, R.: A review of surface ozone background levels and trends, Atmos. Environ., 38, 3431–3442,
- https://doi.org/10.1016/j.atmosenv.2004.03.030, 2004.
- Volz, A. and Kley, D.: Evaluation of the Montsouris series of ozone measurements made in the nineteenth
- century, Nature, 332, 240–242, https://doi.org/10.1038/332240a0, 1988.

- Wang, F. T., Zhang, K., Xue, J., Huang, L., Wang, Y. J., Chen, H., Wang, S. Y., Fu, J. S., and Li, L.:
- Understanding regional background ozone by multiple methods: A case study in the Shandong region,
- China, 2018–2020, J. Geophys. Res.-Atmos., 127, e2022JD036809,
- https://doi.org/10.1029/2022JD036809, 2022a.
- Wang, H., Jacob, D. J., Le Sager, P., Streets, D. G., Park, R. J., Gilliland, A. B., and van Donkelaar, A.:
- Surface ozone background in the United States: Canadian and Mexican pollution influences, Atmos.
- Environ., 43, 1310–1319, https://doi.org/10.1016/j.atmosenv.2008.11.036, 2009a.
- Wang, T., Xue, L., Feng, Z., Dai, J., Zhang, Y., and Tan, Y.: Ground-level ozone pollution in China: a
- synthesis of recent findings on influencing factors and impacts, Environ. Res. Lett., 17, 063003,
- https://doi.org/10.1088/1748-9326/ac69fe, 2022b.
- Wang, T., Wei, X. L., Ding, A. J., Poon, C. N., Lam, K. S., Li, Y. S., Chan, L. Y., and Anson, M.:
- Increasing surface ozone concentrations in the background atmosphere of southern China, 1994–2007,
- Atmos. Chem. Phys., 9, 6217–6227, https://doi.org/10.5194/acp-9-6217-2009, 2009b.
- Wang, Y., Zhang, Y., Hao, J., and Luo, M.: Seasonal and spatial variability of surface ozone over China:
- contributions from background and domestic pollution, Atmos. Chem. Phys., 11, 3511–3525,
- https://doi.org/10.5194/acp-11-3511-2011, 2011.
- Wang, Y., Zhao, Y., Liu, Y., Jiang, Y., Zheng, B., Xing, J., Liu, Y., Wang, S., and Nielsen, C. P.: Sustained
- emission reductions have restrained the ozone pollution over China, Nat. Geosci., 16, 967–974,
- https://doi.org/10.1038/s41561-023-01284-2, 2023.
- West, J. J. and Fiore, A. M.: Management of tropospheric ozone by reducing methane emissions, Environ.
- Sci. Technol., 39, 4685–4691, https://doi.org/10.1021/es048629f, 2005.
- Wilson, R. C., Fleming, Z. L., Monks, P. S., Clain, G., Henne, S., Konovalov, I. B., Szopa, S., and Menut,
- L.: Have primary emission reduction measures reduced ozone across Europe? An analysis of European
- rural background ozone trends 1996–2005, Atmos. Chem. Phys., 12, 437–454,
- https://doi.org/10.5194/acp-12-437-2012, 2012.
- Wu, L., Xue, L., and Wang, W.: Review on the observation-based methods for ozone air pollution
- research, Journal of Earth Environment, 8, 479–491, https://doi.org/10.7515/JEE201706001, 2017.

- Wu, S., Mickley, L. J., Jacob, D. J., Rind, D., and Streets, D. G.: Effects of 2000–2050 changes in climate
- and emissions on global tropospheric ozone and the policy-relevant background surface ozone in the
- United States, J. Geophys. Res.-Atmos., 113, 1–12, https://doi.org/10.1029/2007JD009639, 2008.
- Xie, M., Shu, L., Wang, T., Liu, Q., Gao, D., Li, S., Zhuang, B., Han, Y., Li, M., and Chen, P.: Natural
- emissions under future climate condition and their effects on surface ozone in the Yangtze River Delta
- region, China, Atmos. Environ., 150, 162–180, https://doi.org/10.1016/j.atmosenv.2016.11.053, 2017.
- Xie, W., Xing, Q., Xie, D., Wu, X., Hu, S., and Xu, W.: Pollution characteristics of ozone and its
- precursors in background region of Hainan Province, Environm. Sci., 43, 5407-5420,
- https://doi.org/10.13227/j.hjkx.202201027, 2022.
- Xu, J., Ma, J. Z., Zhang, X. L., Xu, X. B., Xu, X. F., Lin, W. L., Wang, Y., Meng, W., and Ma, Z. Q.:
- Measurements of ozone and its precursors in Beijing during summertime: Impact of urban plumes on
- ozone pollution in downwind rural areas, Atmos. Chem. Phys., 11, 12241–12252,
- https://doi.org/10.5194/acp-11-12241-2011, 2011.
- Xu, W., Lin, W., Xu, X., Tang, J., Huang, J., Wu, H., and Zhang, X.: Long-term trends of surface ozone
- and its influencing factors at the Mt Waliguan GAW station, China Part 1: Overall trends and
- characteristics, Atmos. Chem. Phys., 16, 6191–6205, https://doi.org/10.5194/acp-16-6191-2016, 2016.
- Xu, W., Xu, X., Lin, M., Lin, W., Tarasick, D., Tang, J., Ma, J., and Zheng, X.: Long-term trends of
- surface ozone and its influencing factors at the Mt Waliguan GAW station, China Part 2: The roles
- of anthropogenic emissions and climate variability, Atmos. Chem. Phys., 18, 773–798,
- https://doi.org/10.5194/acp-18-773-2018, 2018.
- Xu, X.: Recent advances in studies of ozone pollution and impacts in China: a short review, Curr. Opin.
- Env. Sci. Hl., 19, 100225, https://doi.org/10.1016/j.coesh.2020.100225, 2021.
- Xu, X., Zhang, T., and Su, Y.: Temporal variations and trend of ground-level ozone based on long-term
- measurements in Windsor, Canada, Atmos. Chem. Phys., 19, 7335–7345, https://doi.org/10.5194/acp-
- 19-7335-2019, 2019.
- Xu, X., Lin, W., Xu, W., Jin, J., Wang, Y., Zhang, G., Zhang, X., Ma, Z., Dong, Y., Ma, Q., Yu, D., Li, Z.,
- Wang, D., and Zhao, H.: Long-term changes of regional ozone in China: Implications for human health

- and ecosystem impacts, Elementa-Sci. Anthrop., 8, 13, https://doi.org/10.1525/elementa.409, 2020.
- Xue, L. K., Wang, T., Zhang, J. M., Zhang, X. C., Deliger, Poon, C. N., Ding, A. J., Zhou, X. H., Wu, W.
- S., Tang, J., Zhang, Q. Z., and Wang, W. X.: Source of surface ozone and reactive nitrogen speciation
- at Mount Waliguan in western China: New insights from the 2006 summer study, J. Geophys. Res.-
- Atmos., 116, 1–12, https://doi.org/10.1029/2010JD014735, 2011.
- Yamaji, K., Ohara, T., Uno, I., Tanimoto, H., Kurokawa, J., and Akimoto, H.: Analysis of the seasonal
- variation of ozone in the boundary layer in East Asia using the Community Multi-scale Air Quality
- model: What controls surface ozone levels over Japan?, Atmos. Environ., 40, 1856–1868,
- https://doi.org/10.1016/j.atmosenv.2005.10.067, 2006.
- Yan, Q., Wang, Y., Cheng, Y., and Li, J.: Summertime clean-background ozone concentrations derived
- from ozone precursor relationships are lower than previous estimates in the southeast United States,
- Environ. Sci. Technol., 55, 12852–12861, https://doi.org/10.1021/acs.est.1c03035, 2021.
- Yang, L., Luo, H., Yuan, Z., Zheng, J., Huang, Z., Li, C., Lin, X., Louie, P. K. K., Chen, D., and Bian, Y.:
- Quantitative impacts of meteorology and precursor emission changes on the long-term trend of
- ambient ozone over the Pearl River Delta, China, and implications for ozone control strategy, Atmos.
- Chem. Phys., 19, 12901–12916, https://doi.org/10.5194/acp-19-12901-2019, 2019.
- Ye, X., Zhang, L., Wang, X., Lu, X., Jiang, Z., Lu, N., Li, D., and Xu, J.: Spatial and temporal variations
- of surface background ozone in China analyzed with the grid-stretching capability of GEOS-Chem
- high performance, Sci. Total Environ., 914, 169909, https://doi.org/10.1016/j.scitotenv.2024.169909,
- 2024.
- Yeo, M. J. and Kim, Y. P.: Long-term trends of surface ozone in Korea, J. Clean. Prod., 294, 125352,
- https://doi.org/10.1016/j.jclepro.2020.125352, 2021.
- Zeng, P., Lyu, X. P., Guo, H., Cheng, H. R., Jiang, F., Pan, W. Z., Wang, Z. W., Liang, S. W., and Hu, Y.
- Q:: Causes of ozone pollution in summer in Wuhan, central China, Environ. Pollut., 241, 852–861,
- https://doi.org/10.1016/j.envpol.2018.05.042, 2018.
- Zhang, L., Jacob, D. J., Downey, N. V., Wood, D. A., Blewitt, D., Carouge, C. C., van Donkelaar, A.,
- Jones, D. B. A., Murray, L. T., and Wang, Y.: Improved estimate of the policy-relevant background

ozone in the United States using the GEOS-Chem global model with 1/2°×2/3° horizontal resolution 1396 North America, Environ., 45, 6769–6776, over Atmos. 1397 https://doi.org/10.1016/j.atmosenv.2011.07.054, 2011. 1398 Zhang, Y., Jin, J., Yan, P., Tang, J., Fang, S., Lin, W., Lou, M., Liang, M., Zhou, Q., Jing, J., Li, Y., Jia, 1399 X., and Lyu, S.: Long-term variations of major atmospheric compositions observed at the background 1400 stations in three key areas of China, Adv. Clim. Change Res., 11, 370-380, 1401 https://doi.org/10.1016/j.accre.2020.11.005, 2020. 1402 Zohdirad, H., Montazeri Namin, M., Ashrafi, K., Aksoyoglu, S., and Prévôt, A. S. H.: Temporal 1403 variations, regional contribution, and cluster analyses of ozone and NO<sub>x</sub> in a middle eastern megacity 1404 during summertime over 2017-2019, Environ. Sci. Pollut. Res., 29, 16233-16249,

https://doi.org/10.1007/s11356-021-14923-1, 2022.

1405