# Peer review of "A comprehensive review of tropospheric background"

_EGUsphere, 2025_

## Author Comment (AC1)

**Responses to Reviewers' comments**

To the esteemed Editor and Reviewers,

We sincerely appreciate the reviewers' time and effort in evaluating our manuscript. Their thoughtful comments and constructive suggestions have been extremely helpful in improving our work. Through careful consideration of all concerns and implementation of the reviewers' recommendations, we believe the revised manuscript has been significantly strengthened.

Please find below our detailed point-by-point responses to the reviewer's comments:

Reviewers' comments are in black.

Author's responses are in blue.

Changes in the manuscript are in red.

Yours Sincerely,

Weihua Chen

On behalf of the authors

**Response to Reviewer 1**

**General comments:**

1.  Chen et al. review the methods for estimating background ozone concentrations, which determine the maximum achievable reduction in ozone pollution through mitigation of local anthropogenic emissions, with a specific focus on background ozone in China. The paper is informative and organized, but it is overly long in some sections.

Response: We thank the reviewer for the positive comments. We have revised the manuscript according to your comments.

2.  It is also important for the authors to say why this review is needed when there are so many other similar recent papers reviewing ozone in China (Liu et al., 2020; Lu et al., 2019; Sahu et al., 2021; Wang et al., 2022; Xu, 2021). I think the key is that this paper analyzes background ozone specifically, but that concept is also addressed quite thoroughly in most papers reviewing drivers of ozone concentrations and trends.

Response: We thank the reviewer for this important comment regarding the novelty of our review. We acknowledge that several recent review papers (e.g., Liu et al., 2020; Lu et al., 2019; Sahu et al., 2021; Wang et al., 2022b; Xu, 2021) have substantially advanced our understanding of $O_3$ pollution in China, including its spatiotemporal characteristics, chemical formation regimes, meteorological drivers, and anthropogenic influences. While some of these reviews have also mentioned the concept of background $O_3$, typically in the context of external contributions or policy challenges, none has provided a systematic and dedicated synthesis on this topic.

At the same time, a growing body of original research has specifically addressed background $O_3$ in China, including its quantification, natural and non-local sources, and policy challenges (Sahu et al., 2021; Wang et al., 2022b; Chen et al., 2022; Wang et al.,

2011). However, these findings remain fragmented and have not been comprehensively reviewed.

Our manuscript aims to fill this gap by: (1) systematically reviewing the evolution of the background $O_3$ concept; (2) providing a comparative assessment of the methods employed to estimate background $O_3$; and (3) synthesizing the spatiotemporal patterns of background $O_3$ in China within a broader global context.

Because background $O_3$ represents the effective lower bound of $O_3$ pollution control, a focused and structured review is both timely and necessary to inform scientific understanding and air quality management.

We have clarified this motivation in Lines 79–94 in the revised manuscript:

Despite its importance, understanding of background $O_3$ remains constrained by inconsistent definitions, diverse estimation methods, and limited regional assessments.

While several recent reviews have touched on background $O_3$ in the broader context of

$O_3$ pollution (Lu et al., 2019b; Liu et al., 2020; Sahu et al., 2021; Wang et al., 2022b;

Xu, 2021), and a few studies have provided quantitative estimates and source apportionment of background $O_3$ in China (Sahu et al., 2021; Wang et al., 2022b; Chen et al., 2022; Wang et al., 2011), no dedicated, methodologically focused synthesis is available.

This study addresses that gap by: (i) systematically reviewing the evolution of the background $O_3$ concept; (ii) providing a comparative assessment of the methods (i.e., in situ measurement, statistical analysis, numerical modeling, and integrated methods)

employed to estimate background $O_3$; and (iii) synthesizing the spatiotemporal patterns of background $O_3$ in China within a broader global context, examining the spatial and temporal variations of background $O_3$ across China using publicly available data.

Finally, we identify key knowledge gaps and propose future research priorities to advance understanding and inform effective policy responses. As background $O_3$

increasingly shapes the $O_3$ landscape, this review provides timely insight into the scientific and regulatory frontiers of $O_3$ pollution control both in China and globally.

3. I would also suggest that the authors significantly reduce the length of Section 4, which is quite long-winded and repetitive.

Response: We have substantially revised this section to improve clarity and conciseness. In particular, we streamlined Section 4.2 (Statistical analysis estimation)

by: (1) removing overly detailed descriptions of individual sub-method (e.g., PCA, K- means, percentile, etc.) of statistical analysis estimation; (2) consolidated repetitive statement on assumptions and limitations; (3) merging overlapping case studies to emphasize representative findings rather than providing an exhaustive list; (4)

relocating the discussion of methodological advantages into Figure 3, thereby avoiding redundancy across subsections.

As a result, the length of Section 4.2 has been reduced by approximately 30–40%, and the revised text is now more concise, readable, and focused while preserving the essential technical content.

4.    Section 5.2 and Figure 5 are the most interesting pieces of the review, and the discussion seemed too short. Why is the integrated method a much smaller range? Is there regional variability in the background ozone estimated from the integrated method?

Should we "trust" the results of the integrated method above the other methods and, if so, why? I think this information is discussed more in Section 4.4, but I would move this information to the section where Figure 5 is discussed.

Response: We sincerely thank the reviewer for highlighting the importance of

Section 5.2 and Figure 5, and for providing valuable suggestions that helped us refine our discussion of the integrated method.

In the revised manuscript, we made the following substantive changes:

(1) Explaining the narrower range. We expanded Section 5.2 to clarify that the narrower range of background $O_3$ concentrations estimates obtained from the integrated methods does not indicate oversimplification but rather reflects its strength in reconciling model structural consistency with observational variability. At the same time, the limited number of integrated studies currently available in China may contribute to the relatively low variability observed, highlighting the need for further validation. International applications (Dolwick et al., 2015; Skipper et al., 2021;

Hosseinpour et al., 2024) provide supporting evidence for its robustness.

(2) Addressing reliability. We clarified that no single method can be considered definitive. Each carries inherent limitations. Instead of replacing other methods, the integrated methods serve as complementary synthesis framework that balances empirical realism with generalizability. Its ability to reduce uncertainty enhances its value for both scientific interpretation and policy-relevant applications.

These revisions (Lines 589–606) strengthen the connection between Section 5.2 and Figure 5, and improve the coherence of the narrative by situating key discussions within the section most relevant to the integrated method:

In contrast, the integrated methods – combining in situ observation, statistical analysis, and numerical results – yield the narrowest range (32–37 ppb), with the value of $34.5 \pm 1.6$ ppb. This narrow range reflects their strength in reconciling the structural consistency of models with real-world variability, rather than oversimplification. By harmonizing data sources, integrated methods reduce methodological noise and yield more robust, policy-relevant estimates. The limited number of applications, however, may also contribute to the observed low variability. Although studies in China remain scarce, international applications underscore their potential. For instance, Skipper et al. (2021) showed that incorporating spatial and temporal bias corrections improved the consistency of model-derived background $O_3$ estimates by 28% relative to unadjusted models. Similarly, Hosseinpour et al. (2024) demonstrated that a random forest machine learning (RF-ML) algorithm integrating multiple data sources with nonlinear feature analysis produced background $O_3$ estimates most consistent with in situ observations for correcting air quality model simulations and outperformed the original CAMx model, multivariate regression, and other ML algorithms. Collectively, these studies highlight the value of integrated methods in producing consistent estimates, particularly for regulatory applications and long-term trend assessments. Nevertheless, further validation is needed to determine whether the observed low variability reflects true methodological robustness or limited sampling. Importantly, no single method is definitive. Each carries inherent assumptions. Integrated methods therefore provide a complementary framework that balances empirical realism with generalizability.

5. In addition, I would expect the paper to have more discussion of the seasonality of background ozone, which is significant in a region like China (Chen et al., 2023).

Response: We thank the reviewer for highlighting the importance of seasonal variability in background $O_3$, especially given China's large climatic and geographic heterogeneity. We have added a subsection to discuss the seasonal variation of background $O_3$ in Lines 687–719 of the revised manuscript.

**5.4 Seasonal variation of background ozone in China**

Figure 7 illustrates the seasonal variations in mean background $O_3$ concentrations across China and its seven subregions during 1994–2020. Nationally, background $O_3$ exhibits pronounced seasonality, with comparable peaks in spring ($47.2 \pm 10.6$ ppb) and summer ($47.3 \pm 15.4$ ppb), and a pronounced minimum in winter ($33.2 \pm 9.8$ ppb).

Regional patterns reveal clear differences in seasonal maxima. In Southwest China (SWC) and Northeast China (NEC), peaks occurred in spring ($52.1 \pm 9.9$ ppb and $38.8 \pm 4.4$ ppb, respectively), largely driven by stratosphere–troposphere exchange (STE) and enhanced downward transport over elevated terrain, and also influenced by prevailing winds that transport $NO_x$ and VOCs from Southeast Asia and other regions into these areas (Liu et al., 2019; Lu et al., 2019a; Xu et al., 2018; Wang et al. 2011; Ye et al., 2024). In contrast, North China (NC), Northwest China (NWC), and East China (EC) recorded summer maxima ($56.8 \pm 10.8$, $55.0 \pm 8.5$, and $48.3 \pm 16.9$ ppb, respectively), consistent with the influence of the East Asian Summer Monsoon (EASM), which enhances precursor inflow and stimulates photochemical $O_3$ formation under high temperatures and intense solar radiation (Gao et al., 2005; Liu et al., 2019, 2021; He et al., 2021). South China (SC) and Central China (CC) reached their highest levels in autumn ($46.9 \pm 10.4$ and $43.0 \pm 14.2$ ppb, respectively), likely reflecting inland pollutant transport by northeasterly winds combined with favorable sunlight conditions (Xie et al., 2022; Shen et al., 2019; Luo et al., 2019).

Seasonal minima also varied by region. Winter lows were observed in Northeast China (NEC, $24.5 \pm 3.6$ ppb), North China (NC, $24.9 \pm 5.2$ ppb), and East China (EC, $25.2 \pm 8.1$ ppb), reflecting weak photochemistry under low temperatures and reduced solar radiation. In contrast, South China (SC, $24.8 \pm 5.0$ ppb) and Central China (CC, $28.7 \pm 10.0$ ppb) exhibited summer minima, attributable to frequent precipitation and high humidity suppressing $O_3$ production. Southwest China (SWC) maintained persistently low levels in both summer ($31.0 \pm 8.2$ ppb) and autumn ($31.0 \pm 4.6$ ppb), whereas Northwest China (NWC) showed relatively lower concentrations in autumn ($41.8 \pm 8.9$ ppb) and winter ($41.9 \pm 5.1$ ppb).

In summary, the seasonal cycle of background $O_3$ in China is shaped by the interplay of regional meteorology and precursor emissions, while vertical exchange and interregional transport further modulate seasonal peaks and troughs across regions.

[Figure]

**Figure 7: Seasonal variations in mean background O₃ concentrations across seven regions of China during 1994–2020. All data sources are compiled and summarized in Table S1. The values of "n =" indicates the number of individual data records or assembly estimates used in the analysis for each region and season.**

6. Finally, I suggest adding the following citations from potentially relevant studies. Han et al. (2019) quantify the influence of foreign ozone on East Asia using GEOS-Chem simulations. Colombi et al. (2023) found that subsidence of elevated free troposphere ozone drove surface background ozone in East Asia in May to exceed 50 ppb, which is much higher than the background values in North America and Europe. Wang et al. (2022) showed that the high free tropospheric ozone in East Asia has been increasing in the last two decades, which may lead to increased background ozone. Ma et al. (2025) diagnose causes of high free tropospheric ozone. Luo et al. (2024) also discuss STE as a source of background ozone in the North China Plain. There may be many other papers I am missing, but I leave the decision of whether to add these citations up to the authors' discretion.

Response: We sincerely thank the reviewer for suggesting these highly relevant and up-to-date references, which substantially strengthen our discussion of the sources and long-term trends of background $O_3$ in East Asia and China. We have carefully reviewed each of the recommended studies and have incorporated them into Sections

5.5 of the revised manuscript.

Lines 738–743:

Several studies have shown that the influx of $O_3$ and its precursors from other regions, including Southeast Asia, Europe, North America, India, and the Middle East, can elevate background $O_3$ concentration in China by 2–15 ppb (Han et al., 2019; Wang et al., 2011; Wang et al., 2022b; Li et al., 2014; Ni et al., 2018). This influence is particularly pronounced during specific seasons when atmospheric circulation facilitates the transboundary transport of atmospheric pollutants (Colombi et al., 2023;

Ma et al., 2025; Ni et al., 2018; Sahu et al., 2021; Ye et al., 2024).

Lines 747–749:

Wang et al. (2022b) also found that emissions outside the Yangtze River Delta regions contributed as much as 63% to $O_3$ pollution within the region.

Lines 753–758:

This process is most active during spring, when stratospheric $O_3$ is transported downward into the troposphere (Ding and Wang, 2006; Lu et al., 2019a; Ma et al., 2025;

Xu et al., 2018). Wang et al. (2011) estimated that STE contributes approximately 7 ppb to background $O_3$ concentrations in northern China during the spring season. Luo et al.

(2024) further revealed that STE contributed an average of 9.6 ppb to surface $O_3$ over the North China Plain during 19–20 May 2019.

7.   Overall, I think this review is potentially useful, but would benefit from shortening

Section 4 and more clearly stating why this review is different and necessary given the many other reviews of ozone in China from the last 5 years.

Response: We thank the reviewer for the positive overall assessment and constructive suggestions. In line with the recommendation, we have shortened Section

4 while improving clarity and readability. In addition, we have strengthened the

Introduction to more explicitly highlight the novelty of this review relative to recent studies on $O_3$ in China. These revisions are detailed in our response to comments #2

and #3.

**Specific comments:**

8.     I suggest adding a sentence that clearly describes the concept of background ozone to the Introduction of the paper? For example, around line 68 in the introduction,

I think a sentence is needed that says something like: "Background surface ozone is defined as the ozone that would be present in a given region in the absence of local anthropogenic emissions, and can be influenced by local natural factors or distant anthropogenic and natural factors." It is stated in the abstract, but should be defined again in the introduction.

Response: We thank the reviewer for this thoughtful suggestion and fully agree that an explicit early definition of background $O_3$ is essential to guide readers'

understanding of the scope and focus of the review.

Accordingly, we have added the following sentence in the Introduction (Lines 41–

44) of the revised manuscript:

Background $O_3$ refers to the $O_3$ concentration present in the absence of local anthropogenic precursor emissions. It originates from a variety of natural and non-local processes, including methane ($CH_4$) oxidation, stratosphere–troposphere exchange (STE), vegetation, soil, lightning, wildfires and long-range pollutants transport, as shown in Fig. 1 (Dolwick et al., 2015; Thompson, 2019).

9.     Figure 1 is a nice visualization. I like how the beaker implies chemical reactions occurring in the atmosphere, and shows how anthropogenic emissions add ozone on top of the background.

Response: We sincerely thank the reviewer for the positive feedback on Figure 1.

We are pleased that the design successfully conveys both the chemical processes in the atmosphere (as symbolized by the beaker) and the additive effect of anthropogenic emissions on top of the background $O_3$.

10.     Line 163: The conversion from $\mu g\ L^{-3}$ to ppb is strange to me. Why would you assume a temperature of 0˚C? How big of a difference does it make if you assume a more reasonable temperature and pressure rather than STP? Could you use meteorological data to assume a more accurate and precise temperature and pressure?

Response: We thank the reviewer for this important and constructive comment. In the original manuscript, the conversion from mass concentration ($\mu g \cdot m^{-3}$) to volume mixing ratios (ppb) was based on standard temperature and pressure (STP, 0 °C and

101.325 kPa) following the original "Ambient Air Quality Standards" (GB 3095–2012)

issued by the Ministry of Ecology and Environment of China. However, as the reviewer rightly notes, this assumption does not represent the typical meteorological conditions in China and differs from internationally recognized reference states, such as 25 °C in the United States (U.S. EPA, 2011), 20 °C in the European Union (European Parliament and Council, 2008).

To address this, we have revised our calculations using a reference state of 25 °C

(298.15 K) and 101.325 kPa, consistent with the 2018 amendment to GB 3095–2012

and international practice. This corresponds to a molar volume of 24.5 $L \cdot mol^{-1}$. We further validate this adjustment, we performed an analysis using meteorological data from over 400 national monitoring sites across China, spanning four representative months (January, April, July, and October 2022). Applying the ideal gas law ($V_m$ =

RT/P), we found that 95% of the calculated molar volumes under actual ambient conditions fall within the range of 20.16–30.4 $L \cdot mol^{-1}$, with the revised value of 24.5

$L \cdot mol^{-1}$ well within this representative range. Therefore, this updated assumption better reflects the climatological conditions in most Chinese regions and improves comparability with international datasets and modeling studies.

All relevant conversion results in the manuscript have been updated accordingly.

Since these adjustments affect only numerical values and do not alter the conceptual framework or conclusions, we have not listed each recalculation in detail. The updated description and formula are now provided in Lines 138–148 of the revised manuscript:

To ensure consistency with international standard and comparability with global datasets, unit conversions were performed using Eq. (1):

$$ppb = \left(\frac{24.5 \text{ L mol}^{-1}}{48 \text{ g mol}^{-1}}\right) \times (\mu g \text{ m}^{-3}),\qquad\qquad(1)$$

Where 48 $\text{g mol}^{-1}$ is the molar mass of $O_3$ and 24.5 $\text{L mol}^{-1}$ is the molar volume of an ideal gas under the reference conditions of 25 °C and 1013.25 hPa, as specified in the 2018 amendment to China's Ambient Air Quality Standards (GB 3095–

2012) issued by the Ministry of Ecology and Environment (https://www.mee.gov.cn/gkml/sthjbgw/sthjbgg/201808/t20180815_451398.htm).

These reference conditions are consistent with international practices, such as 25 °C in the U.S. (U.S. EPA, 2011), 20 °C in the European Union (European Parliament and

Council, 2008) and better reflect typical meteorological conditions across most regions of China.

11.  Figure 3 is pretty wordy and not easy to absorb the information presented. I

suggest significantly revising this figure to make it easier to digest and more visually appealing. At the very least, the authors should remove unnecessary words (e.g.,

"suitable" is over-used).

Response: We sincerely thank the reviewer for this valuable suggestion. We agree that the original Figure 3 was overly text-heavy and could hinder comprehension. To address this, we have substantially revised the figure by: (1) removing redundant and non-essential wording, including the overused term "suitable"; and (2) condensing methodological descriptions into concise, keyword-based annotations that highlight the essential features and comparative distinctions of each approach.

The revised Figure 3 is now more concise, visually appealing, and easier to interpret:

[Figure]

**Figure 3: Summary of the advantages, limitations, and applicability of different estimation methods for**
**background O$_3$.**

12.    Figure 4: Could you please make the shading of each region be a color that represents the magnitude of the background ozone concentration? For example, a color bar that ranges from 25 to 50 ppb where lower background ozone is a lighter color and higher background is a darker color. It is hard to view the spatial variability of background ozone from the bar charts.

Response: We thank the reviewer for this insightful suggestion. We agree that the original bar chart format limited the visualization of spatial variability. Accordingly, we have redesigned Figure 4 as a regionally shaded map, where background O₃

concentrations are represented by a discrete color scale ranging from 24 to 49 ppb (lighter colors indicate lower values and darker colors higher values). This revision greatly improves the clarity and intuitiveness of spatial patterns. The updated figure, caption, and corresponding text have been incorporated into the revised manuscript (Lines 559–563):

[Figure]

**Figure 4: Spatial distribution of background O₃ concentrations (1994–2020) across various regions of China.**
**The locations of 33 background monitoring stations are indicated with red dots. The seven regions include**
**Northeast China (NEC), North China (NC), East China (EC), Central China (CC), Northwest China (NWC),**
**Southwest China (SWC), and South China (SC).**

13.    Figure 5: where are the estimates of background ozone from numerical modeling, statistical methods, and integrated methods coming from? I assume it is from

Table S1, but there is no citation or reference to table S1 in figure caption. Is n= at the bottom of each box referring to the number of studies compiled or the number of measurements? Overall, this caption should be more descriptive.

Response: We thank the reviewer for this constructive comment. We confirm that the estimates in Figure 5 were compiled from Table S1 in the Supplementary

Information. In the revised manuscript, the figure caption has been updated to explicitly reference Table S1 as the data source and to clarify that "n =" denotes the number of individual data records or compiled estimates (i.e., regional mean values reported in peer-reviewed studies), rather than raw measurements or publications. These revisions (Lines 637–639) make the caption more descriptive and improve clarity for readers:

Figure 5: Estimated regional average background $O_3$ concentrations in China from

1994 to 2020 based on multiple methods. All data sources are compiled and summarized in Table S1. The values of "n =" below each box indicate the number of individual data records used in the analysis for each method category.

14.    Section 5.3 and Figure 6: Why is NC the only region that has two separate lines showing the trends in background ozone?

Response: We sincerely thank the reviewer for this thoughtful observation. The

North China (NC) region is shown with two separate trend lines in Figure 6 because of methodological differences among studies. Specifically, Ma et al. (2016) reported a long-term series based on MDA8 $O_3$ concentrations using in situ filtering, while most other studies in NC used hourly averages over shorter periods with different estimation methods. As a result, the MDA8 $O_3$ – based estimates are systematically higher than the hourly average – estimates, and combining them would introduce bias. To preserve methodological consistency and ensure clarity in trend interpretation, we therefore display the two datasets separately. This distinction highlights the contrast between long-term MDA8 $O_3$ trends and shorter-term hourly average studies. We have clarified it in Lines 659–666 of the revised manuscript:

Notably, for North China (NC), two separate trend lines are presented in Figure

6(c), reflecting methodological differences among studies: Ma et al. (2016) provided a long-term record using MDA8 $O_3$ concentrations filtered from in situ observations, while most other studies used hourly averages over shorter or discontinuous periods.

Since MDA8 $O_3$-based estimates are inherently higher than hourly means, aggregating them would bias trend interpretation. Therefore, separate presentation ensures consistency. Furthermore, MDA8 $O_3$ records are scarce elsewhere (typically fewer than four data points), precluding dual-trend comparison.

15.     Same comment for Figure 7 as made for Figure 4. Please consider shading the regions according to the background ozone concentration

Response: We have replotted Figure 7 (now is Figure 8) as a regionally shaded map in Line 816 of the revised manuscript:

[Figure]

**Figure 8: Average background O₃ concentrations in the U.S., Canada, Europe, South Korea, Japan, and China.**

16. The conclusions section is informative and written well.

Response: We sincerely thank the reviewer for their positive and encouraging comments regarding the conclusions section.

**Technical corrections:**

17. Lines 8–10: The first sentence of the abstract is worded in a confusing way. I suggest: Background ozone (O₃) refers to baseline O₃ concentrations in the absence of local anthropogenic emissions, critical for understanding and mitigating tropospheric O₃ pollution.

Response: We thank the reviewer for the helpful suggestion. We have revised the first sentence of the abstract (Lines 8–9) to make the definition clearer and more concise:

Background ozone (O₃) represents the baseline concentrations in the absence of local anthropogenic emissions and is critical for understanding and mitigating tropospheric $O_3$ pollution.

**Response to Reviewer 2**

**Major comments:**

1.      Chen et al. present a comprehensive review of definitions and methods for estimating tropospheric background ozone concentration, with a specific focus on China. The manuscript is well-structured and demonstrates considerable effort in literature compilation. However, I have several major concerns that should be addressed before the manuscript can be considered for publication.

Response: We sincerely thank the reviewer for the constructive feedback and recognizing the value of our literature review. Detailed responses to each comment are provided below.

2.   For Section 3, the authors discuss multiple definitions, including USBO, PRBO, RBO, and NBO. While the historical evolution is clearly present, the distinctions between these items may be difficult for readers to follow. I suggest adding a summary table outlining the key characteristics of each definition, such as name, included and excluded sources, and typical applications. Alternatively, the authors could consider incorporating these details into Figure 2.

Response: We thank the reviewer for this constructive suggestion. To improve clarity in distinguishing among NBO, USBO, PRBO, and RBO, we have substantially revised Figure 2. The updated version explicitly highlights the key elements of each concept, presented in a side-by-side format that links definitions (left box) with their corresponding features and applications (right box). This integrated design captures the intent of a summary table while maintaining the visual continuity of the historical evolution, making the distinctions among different concepts clearer and more accessible to readers. The revised Figure 2 is provided in Section 3 of the revised manuscript:

[Figure]

**Figure 2: Historical evolution of background O₃ concepts: definitions (left box) and characteristics (right box).**

3. In Section 4.2, the authors introduce several statistical methods estimating background $O_3$. It would be helpful if the authors could compare the results from these different methods at the same location, using an example to illustrate their differences to highlight the respective advantages and disadvantages of each approach.

Response: We thank the reviewer for this insightful comment. We fully agree that a direct comparison of background $O_3$ estimates derived from different methods at the same location would provide a clearer illustration of their respective strengths and limitations. However, such a systematic comparison is not feasible in the present study because (1) not all methods introduced in Section 4.2 have been applied in China, and those that have (e.g., PCA, TCEQ, temperature-ozone relationship, and nocturnal $O_3$

methods) require different observational datasets with region-specific applicability; and (2) existing studies across China vary in spatial coverage and study periods, making it difficult to apply multiple approaches consistently to the same region.

To address the reviewer's concern, we expanded our discussion by incorporating evidence from studies that have compared multiple methods within the same region.

For example, in Shandong, Wang et al. (2022a) reported that PCA (using ambient $O_3$

alone) yielded background $O_3$ about 20 ppb higher than the TCEQ approach, with seasonal patterns more consistent with background-site observations, whereas the

TCEQ method tended to underestimate background $O_3$ due to residual urban influences.

In the inland southeastern United States, Yan et al. (2021) found systematic differences across three methods ($O_3$-CO-HCHO, 1-$\sigma$ $O_3$-$NO_z$, and percentile-based), reflecting varying sensitivities to anthropogenic signals. Similarly, Chen et al. (2022) revealed that the nocturnal $O_3$ method underestimated background $O_3$ by up to 30% compared with the temperature-ozone relationship method in China during polluted seasons.

We have expanded the discussion in Lines 607–635 of the revised manuscript:

Method-dependent discrepancies underscore the complexity of estimating background $O_3$. Variability arises from differences in input data, model assumptions, and the parameterization of physical and chemical processes (Jaffe et al., 2018; Skipper et al., 2021; Wang et al., 2022a; Yan et al., 2021). For instance, in situ measurement estimation method is directly influenced by local meteorological and emission conditions, whereas the numerical modeling estimation method is subject to uncertainties in simulating processes such as natural emissions, transboundary transport, and photochemical reactions. Ideally, direct comparison of background $O_3$ estimates derived from multiple methods at the same location would clarify their relative strengths and limitations. However, such comparison was not feasible here due to methodological and data constraints. First, the dataset used in this study is limited to China, where only a subset of the methods described in Sect. 4.2 has been applied, each requiring specific datasets and exhibiting region-dependent applicability. Second, background $O_3$ exhibits pronounced spatial and temporal variability, while existing studies often target different subregions and time periods, making consistent co-located comparisons impractical. Despite these challenges, several studies have conducted preliminary intercomparisons within the same region. In Shandong, Wang et al. (2022a) reported that PCA (using ambient $O_3$ alone) yielded background $O_3$ about 20 ppb higher than the TCEQ approach, with seasonal patterns more consistent with background-site observations. The TCEQ method tended to underestimate background $O_3$ because minimum MDA8 $O_3$ values were often influenced by residual urban emissions. In the inland southeastern U.S., Yan et al. (2021) found the $O_3$-CO-HCHO method yielded the lowest estimates (10–15 ppb), the 1-σ $O_3$-$NO_z$ method intermediated values (15–25 ppb), and the 5[th] percentile method the highest (20–30 ppb), likely due to anthropogenic influences in urban downwind regions. Likewise, Chen et al. (2022) revealed that the nocturnal $O_3$ method underestimated background $O_3$ by up to 30% compared with the temperature-ozone relationship method during polluted seasons in China.

Collectively, these studies demonstrate that methodological choices alone can lead to discrepancies of 10–20 ppb in background $O_3$ estimates within the same region. Careful interpretation therefore requires explicit attention to methodological assumptions, data representativeness, and sensitivity to emission influences. Moving forward, the development of harmonized datasets would enable the consistent application of multiple methods at the same regions and time periods, providing more robust intercomparisons and clearer insights into the strengths and limitations of each approach.

4. Also, the manuscript suggests that integrated methods improve the estimation of background $O_3$ by combining various data sources and techniques in section 4.4. How it is unclear how these results compare quantitatively with those from individual methods discussed in the above sections. And how strong are we confident with these integrated results? Providing a concrete example or case study to illustrate the improvement would strengthen this section.

Response: We thank the reviewer for this constructive comment and agree that a quantitative comparison between integrated methods and the individual approaches would strengthen the discussion. At present, however, studies that directly apply both integrated and single-method estimations to the same region and time period are not available, limiting the possibility of a fully consistent side-by-side comparison.

To address this point, we have revised Section 4.4 to highlight quantitative evidence where integrated methods have been evaluated against individual approaches.

For example, Skipper et al. (2021) reported that incorporating spatial and temporal bias corrections improved the consistency of model-derived background $O_3$ estimates by 28%

compared with the unadjusted model. Likewise, Hosseinpour et al. (2024)

demonstrated that a random forest machine learning (RF-ML) algorithm integrating multiple data sources with nonlinear feature analysis produced background $O_3$

estimates most consistent with in situ observations for correcting air quality model simulations and outperformed the original CAMx model, multivariate regression, and other ML algorithms. These case studies provide concrete examples of the advantages of integrated methods.

Regarding confidence, we note that the reliability of integrated methods depends on the representativeness of the underlying observational and modeling datasets and the assumptions embedded in the integration process. Nonetheless, current evaluations consistently indicate that such frameworks reduce systematic biases, improve spatial- temporal coherence, and align more closely with independent in situ estimates. While further validation is needed using side-by-side comparisons with single-method estimates, existing evidence supports their strong potential to deliver more reliable and policy-relevant background $O_3$ estimates.

We have incorporated these clarifications and supporting examples in Lines 589–

606 of the revised manuscript:

In contrast, the integrated methods – combining in situ observation, statistical analysis, and numerical results – yield the narrowest range (32–37 ppb), with the value of 34.5 ± 1.6 ppb. This narrow range reflects their strength in reconciling the structural consistency of models with real-world variability, rather than oversimplification. By harmonizing data sources, integrated methods reduce methodological noise and yield more robust, policy-relevant estimates. The limited number of applications, however, may also contribute to the observed low variability. Although studies in China remain scarce, international applications underscore their potential. For instance, Skipper et al. (2021) showed that incorporating spatial and temporal bias corrections improved the consistency of model-derived background $O_3$ estimates by 28% relative to unadjusted models. Similarly, Hosseinpour et al. (2024) demonstrated that a random forest machine learning (RF-ML) algorithm integrating multiple data sources with nonlinear feature analysis produced background $O_3$ estimates most consistent with in situ observations for correcting air quality model simulations and outperformed the original CAMx model, multivariate regression, and other ML algorithms. Collectively, these studies highlight the value of integrated methods in producing consistent estimates, particularly for regulatory applications and long-term trend assessments. Nevertheless, further validation is needed to determine whether the observed low variability reflects true methodological robustness or limited sampling. Importantly, no single method is definitive. Each carries inherent assumptions. Integrated methods therefore provide a complementary framework that balances empirical realism with generalizability.

5. The meta-analysis is a key strength of this manuscript, but details on how the data were compiled and interpreted are limited. Clarifying the data processing criteria and handling of uncertainties would enhance this part of the work.

Response: We thank the reviewer for highlighting the importance of clarifying data processing criteria and the treatment of uncertainties in our meta-analysis. We have substantially revised Section 2 to provide a more transparent description of how the data were compiled, screened, and harmonized.

Lines 114–121:

[revised manuscript text omitted]